

**Large ensemble simulations of the North American and Greenland ice**
**sheets at the Last Glacial Maximum with a coupled atmospheric general**
**circulation-ice sheet model**
Sam Sherriff-Tadano[1], Ruza Ivanovic[1], Lauren Gregoire[1], Charlotte Lang[2], Niall Gandy[1, 5], Jonathan
Gregory[2,3], Tamsin L. Edwards[4], Oliver Pollard[1] and Robin S. Smith[2]
[1] School of Earth and Environment, University of Leeds, UK
[2] National Centre for Atmospheric Science, University of Reading, UK
[3] Met Office Hadley Centre, Exeter, UK
[4] King's College London, UK
[5] Department of Natural and Built Environment, Sheffield Hallam University, UK
*Correspondence to*: Sam Sherriff-Tadano (tadanosam@gmail.com)
**Abstract**
The Last Glacial Maximum (LGM) is characterised by huge ice sheets covering the Northern Hemisphere, especially over
North America, and by its cold climate. Numerical simulations of the climate and ice sheets of the LGM have been
performed to better understand these systems, however the inherent uncertainty and sensitivity in the simulations to the
selection of model parameters remain uncertain. Here, we perform a 200-member ensemble of simulations of the North
American and Greenland ice sheets and climate of the LGM with an ice sheet-atmosphere-slab ocean coupled model
(FAMOUS-BISICLES) to explore sensitivities of the coupled climate-ice system to 16 uncertain parameters. In the ensemble
of simulations, the global temperature is primarily controlled by the combination of parameters in the large-scale
condensation scheme and the cumulus convection scheme. In simulations with plausible LGM global temperatures, we find
that the albedo parameters have only a small impact on the Greenland ice volume due to the limited area of surface ablation
associated with the cold climate. Instead, the basal sliding law controls the ice volume by affecting ice transport from the
interior to the margin. On the other hand, like the Greenland ice sheet in future climate change, the LGM North American ice
sheet volume is controlled by parameters in the snow and ice albedo scheme. Few of our simulations produce an extensive
North American ice sheet when the global temperature is above 12℃. Based on constraints on the LGM global temperature,
the ice volume and the southern extent of the North American ice sheet, we select 16 acceptable simulations. These
simulations lack the southern extent of ice compared to reconstructions, though show reasonable performance on the ice
sheet configuration and ice streams facing the Baffin Bay and the Arctic Ocean. The strong sensitivities of the North
American ice sheet to albedo at the LGM may imply a potential constraint on the future Greenland ice sheet by constraining
the albedo schemes.



## 1. Introduction

The rise in sea level predicted in the next several centuries associated with increasing greenhouse gases and global warming is one of the largest concerns of society and the climate community. The most recent IPCC WG1 report projects a global mean sea-level rise of more than 3 m under the high end of the increase in radiative forcing (SSP5-8.5) in the next 300 years (IPCC 2021). However, there are still large uncertainties in the predicted sea level rise with the possibility of a much larger magnitude (Edwards et al. 2021). This large uncertainty in the projection of sea-level rise reflects the present limited state of knowledge of several important processes, such as nonlinear behaviours in the ice sheet system (Gregoire et al. 2012, Abe-Ouchi et al. 2013, Golledge et al. 2019) and interactions of the climate and the ice sheets, which are expressed in climate-ice sheet coupled models (Deconto and Pollard 2016, Golledge et al. 2019, Gregory et al. 2020, Smith et al. 2021). This uncertainty shows the importance of improving our understanding of the ice sheet-climate coupled system and to refine numerical models used for the future projection of climate and sea-level rise.

One method of evaluating climate-ice sheet coupled models and improving understanding of the climate-ice sheet coupled system is to simulate conditions of past periods. In this regard, the Last Glacial Maximum (LGM), which corresponds to approximately 21 thousand years before present (ka BP; Clark et al. 2009, Kageyama et al. 2021), is useful since both climate conditions and the ice sheet configurations are relatively well documented compared to previous periods of glaciation (Tarasov et al. 2012, Kageyama et al. 2021). It has been suggested that the LGM could be used to constrain the climate sensitivity (Tierney et al. 2020), cloud processes (Zhu et al. 2022) and deep ocean circulation (Sherriff-Tadano et al. 2023), implying that understanding this period has the potential to help constrain climate and ice sheet models and future sea level projections. During this period, weaker summer insolation and lower concentrations of greenhouse gases caused the climate to be colder, allowing ice sheets to expand over North America and Northern Europe. As a result, the global climate was colder by 1.7°C to 8.3°C (Holden et al. 2010, Schmittner et al. 2011, Tierney et al. 2020, Paul et al. 2021) and global mean sea level was approximately 120 m lower compared to modern (Clark et al. 2009, Gowan et al. 2021). The mass of the Greenland ice sheet is thought to have been larger by approximately by 2 to 5 m sea level equivalent (SLE) at the LGM (Clark and Mix 2002, Lecavalier et al. 2014, Bradley et al. 2018, Tabone et al. 2018) and of the Antarctic ice sheet by 5.6 to 14.3 m SLE (e.g. Briggs et al. 2014). The Eurasian ice sheet is thought to have attained a volume of 24 m SLE (Hughes et al. 2016), but by much the largest part of the 120 m SLE is attributed to the growth of the North American ice sheet (at least 60 m SLE, e.g. Abe-Ouchi et al. 2015). The position of the margin of the North American ice sheet is constrained reasonably well by geological evidence and this line of evidence is often used to validate the performance of ice sheet models (e.g., Dyke et al. 2002, Clark et al. 2009).

Studies that simulate LGM climate and ice-sheets have primarily treated these components independently using individual numerical models. To investigate the effect of ice sheets on climate, following Manabe and Broccoli (1985), many simulations have been performed and compared their simulations with climate models as part of the Paleoclimate Model Intercomparison Project (PMIP, Braconnot et al. 2007, 2012, Ivanovic et al. 2016, Kageyama et al. 2017). In these simulations, the ice sheet configuration was specified as a boundary condition and they show the important role of the existence of the glacial ice sheets on the climate, affecting surface temperature, precipitation, atmospheric and oceanic circulation (Klockmann et al. 2016, Gregoire et al. 2018, Ivanovic et al. 2018, Sherriff-Tadano et al. 2021). To investigate the effect of climate on ice sheets, simulations of the LGM ice sheets have been performed with ice sheet models. These simulations were performed either as full glacial cycle experiments (e.g. Abe-Ouchi et al. 2007) or perpetual LGM experiments (e.g. Alder and Hostetler 2019). In these experiments, the ice sheet models were forced with climatic conditions based on outputs from general circulation models (Gregoire et al. 2012, Abe-Ouchi et al. 2013, Alder and Hostetler 2019, Niu et al. 2019, Blasco et al. 2021). They showed the critical effects of uncertain climatic conditions and albedo in causing a



large diversity in the simulated ice sheet configuration (Abe-Ouchi et al. 2007, Alder and Hostetler 2019, Niu et al. 2019,
Blasco et al. 2021) together with uncertainties in basal sliding law (Gandy et al. 2020). These studies highlighted the strong
interaction of climate and ice sheets and the importance of performing simulations with climate-ice sheet coupled models to
better understand the coupled system.

Recent efforts in the modelling community in developing coupled climate-ice sheet models (e.g. Gregory et al. 2012, Ziemen
et al. 2014, Roche et al. 2014, Smith et al. 2021) mean that higher complexity coupled climate–ice simulations of the glacial
period than have previously been possible may now be performed. Gregory et al. (2012) performed simulations of an ice
sheet inception over North America with the climate-ice sheet coupled model FAMOUS-Glimmer. They showed the role of
the albedo on the magnitude and speed of the inception. Ziemen et al. (2014) performed simulations of the ice sheet-
atmosphere-ocean with a more complex ice sheet-climate coupled model. Their simulation reproduced the climate and the
ice sheets of the LGM reasonably well, while the southern extent of the North American ice sheet was somewhat smaller
compared to reconstructions. This is partly due to the relatively coarse resolution of the atmospheric model (Ziemen et al.
2014), which means their model underestimated the stationary wave effect that cools the southern extent of the North
American ice sheet and hence underestimates the ice area in that region (Roe and Lindzen 2001, Abe-Ouchi et al. 2007).
Lofverstrom et al. (2015) performed simulations of the North American ice sheet and climate with an atmosphere-ice sheet-
slab ocean coupled model in an idealised framework and showed the importance of interactions between atmospheric
circulation, the Rocky Mountains and the ice sheet in shaping the ice sheet's zonally asymmetric features. Willeit and
Ganopolski (2016) presented simulations of the last glacial cycle with an ice sheet model coupled to an Earth System model
of intermediate complexity and discussed the role of the darkening effect of snow. Quiquet et al. (2021) performed
simulations of the ice sheets and climate of the LGM and the last deglaciation with a coupled climate-ice sheet model. They
managed to reproduce the overall characteristics of the evolution of climate and ice sheets and showed the effects of
modulations in the oceanic circulation.

These previous studies provide very useful insight into the physical interactions within the coupled system, but the inherent
uncertainty and sensitivity in the simulations to the selection of model inputs (including physical parameterisations) remain
untested as in all of these studies a single version of a given model was used. Perturbed parameter ensembles of simulations
are a powerful way to estimate uncertainties originating from particular parameter values in a single model (Murphy et al.
2004, Sanderson 2011, Shiogama et al. 2012). For example, Rougier et al. (2009) analysed results from an ensemble
performed under modern and future climate conditions with an atmosphere-slab ocean coupled general circulation model
(HadSM3) and showed the critical role of entrainment rate in the cumulus cloud scheme and its interaction with large-scale
condensation scheme on global climate. Gregoire et al. (2011) performed an ensemble of simulations with an atmosphere-
ocean coupled general circulation model, FAMOUS, and found that the mid-latitude cloud parameters and sea ice albedo
exert an important influence on global cooling at the LGM. Furthermore, they used their results to identify combinations of
parameter values that optimise model skill in simulating both the pre-industrial and LGM, thus improving model flexibility.
Gandy et al. (2023) recently performed ensemble simulations of the North American ice sheet and climate with an
atmosphere-ice sheet coupled model FAMOUS-Ice (Smith et al. 2021). They showed the importance of ice and snow albedo
in building the ice sheet due to strong summer insolation at the southern margin of the North American ice sheet. In this
study, however, the sea surface temperature and the global temperature were fixed. As a result, the role of clouds on the
climate and the effects of global mean temperature on the ice sheet volume remained unclear.

Here, we perform a large ensemble of simulations of the North American and Greenland ice sheets and climate of the LGM
with a version of the FAMOUS-Ice coupled atmosphere-ice sheet model, which utilises the ice sheet model BISICLES rather



than Glimmer (Method, e.g. Smith et al. 2021).  With this model, we estimate the impact of uncertainty in the choice of
parameter values implemented in the atmosphere and ice sheet components of the model and test the ability of the model to
simulate ice sheets and climates very different from today. The results are evaluated against the LGM global mean
temperature, ice volume and southern extent of the North American ice sheet. Through these experiments, we aim to address
the following questions;
● How do uncertain parameters affect the climate and ice sheets at the LGM?
● Is there a difference in important parameters between the North American and Greenland ice sheets?
● How well are the ice sheets simulated in this experiment,e.g. in terms of North American ice sheet volume, the
southern extent of the North American ice sheet and the position of the ice streams?

The remainder of the paper is structured as follows. Section 2 gives a description of the model, the experimental design and
the integration procedure. Section 3 reports on the results of the large ensemble. Section 4 discusses the results and the effect
of biases in the model. Lastly, section 5 gives the conclusions.

**2. Method**
**2.1 Model**
Our simulations of the climate and ice sheets are performed with the atmosphere-ice sheet-slab ocean coupled model,
FAMOUS-Ice (Smith et al. 2021, Gregory et al. 2020). FAMOUS is a low-resolution version of the atmosphere-ocean
general circulation model (AOGCM) HadCM3; the horizontal resolution is 7.5˚ in longitude and 5˚ in latitude (Smith et al.
2008, 2012). Due to the lower resolution, FAMOUS runs 10 times faster compared to HadCM3, while retaining a reasonable
performance for the modern and the LGM climates (Smith et al. 2008, 2012, Smith and Gregory 2012). Benefitting from
much cheaper computational cost, it is feasible to run multi-millennial simulations (Smith and Gregory 2012) and large
ensembles (Gregoire et al. 2011), as required to meet our objectives.

The latest version of FAMOUS (FAMOUS-ice, Smith et al. 2021) incorporates a downscaling scheme for the calculation of
the surface mass balance (SMB) over ice sheets. In the downscaling scheme, 10 additional vertical tiles are added to better
represent the elevation dependence of surface temperature and downward longwave radiation, following the method first
used in Vizcaino et al. (2013). The downscaled temperature and longwave radiation are then utilised with downward
shortwave radiation to calculate the SMB based on a surface energy budget scheme, together with precipitation from the
original FAMOUS grid. The model also incorporates an updated snow and ice albedo scheme, which accounts for albedo
changes associated with modifications in surface air temperature (*daice*), grain size (*avgr*) and density of the snow (*fsnow*)
(Smith et al. 2021, Table 1). As a result, the atmospheric model reproduces the general pattern of SMB over the modern
Greenland ice sheet reasonably well (van de Wal et al. 2012, Smith et al. 2021) with some overestimating biases in the
elevation of Equilibrium-Line Altitude (ELA; Smith et al., 2021, see also subsection 4.2).

Previous work with FAMOUS-ice used prescribed climatological SSTs and sea-ice instead of an interactive ocean model
(Gregory et al., 2020; Smith et al., 2021, Gandy et al. 2023). In the present study, we use a slab ocean model with the same
horizontal resolution as the atmosphere. Inclusion of a slab ocean model allows the local and global SST and sea-ice  to vary
in response to changes in climate, which in our experiments are caused by modifications in parameters and the advance and
retreat of ice sheets. In the slab ocean model, sea-ice is advected by the climatological monthly surface sea-water velocity of
the HadCM3 pre-industrial control experiment, with sea-ice convergence prevented when the local thickness exceeds 4.0 m.



The local thickness of sea ice evolves due to snowfall, sublimation and melting at the surface, and melting and freezing at
the base in response to heat exchange with the slab ocean. The SST is the temperature of a layer of water 50 m thick, and
evolves in response to surface energy exchange with the atmosphere and heat transport within the slab ocean. Since the slab
ocean does not simulate ocean dynamics, climatological heat transport is prescribed within it as a monthly climatological
field of heat convergence. The heat convergence field is obtained from a calibration experiment (Section 2.2) in which the
model calculates the heat flux necessary to maintain a reference climatological state of SST and sea-ice.
The slab ocean model is essentially the same as described by Williams et al. (2001), where it is used with the HadCM3
AGCM, but the present study is the first to use it with the atmosphere resolution of FAMOUS. For this configuration, grid
boxes which are partly land and partly sea were implemented in the slab ocean, as in the AGCM. In order to prevent unstable
surface temperature feedbacks in coastal grid boxes with small sea fraction, we found that horizontal diffusion of heat in the
slab ocean was needed (diffusivity 10000 $m^2$ $s^{-1}$); unlike the prescribed heat convergence, diffusive heat divergence responds
to the time-dependent slab temperature gradient and thus dissipates local anomalies, but usually it is much smaller than the
heat convergence. In order to prevent local build-up of excessively thick coastal sea ice, we allow horizontal diffusion of sea
ice thickness (diffusivity 5000 $m^2$ $s^{-1}$) when the local thickness exceeds 4.0 m. To improve the reproduction of the reference
sea-ice climatology, we adjusted the coefficients for sea-ice basal melting.
Instead of Glimmer, we use the more complex and computationally demanding BISICLES ice sheet model (Cornford et al.
2013). BISICLES is a vertically integrated ice sheet model, which has been mainly used for simulations of modern and
future Greenland (Lee et al. 2015, Smith et al. 2021b) and Antarctica (Martin et al. 2019, Smith et al. 2021b), and has
recently been applied for reproducing past ice sheets over North America (Matero et al. 2020) and Northern Europe (Gandy
et al. 2018, 2019, 2020). Whereas Glimmer uses the shallow ice approximation, BISICLES applies a L1L2 approximation,
which allows more flexibility in sliding and flowing of the ice sheet especially at the ice shelf area (Cornford et al. 2013). In
addition, the model is capable of changing spatial resolution according to the flow regime of the ice. In this study, a
horizontal resolution of 32 km is chosen, with refinement to 16 km at ice sheet margins. The choice of the resolution was
made based on practical reasons regarding the computational expense. We show that this resolution is adequate for
simulating large-scale glaciers in the northern area of the North American ice sheet (see subsection 4.1).
We utilise a basal drag scheme introduced by Gandy et al. (2019), which explicitly expresses the thermodynamic interaction
of the ice sheets and the underlying till. This scheme combines the Coulomb-friction law and Weertman-friction law
depending on the water pressure in the bedrock sediment (Tsai et al. 2015). The basal drag follows the Weertman law under
cold ice basal temperature and dry bedrock sediment. Under warm ice basal temperature and wet bedrock sediment, the basal
drag follows the Coulomb-friction law. Depending on the depth of till water in the sediment, the friction of ice and bedrock
changes. The depth of the till water is controlled by the balance of basal melting of the ice sheet and a parameter (*drain*) that
controls the vertical till-stored drainage rate. Using this basal scheme in BISICLES simulations, Gandy et al. (2019)
reproduced the features of known ice streams in the LGM British ice sheet.
Changes in ice-sheet geometry, and the subsequent redistribution of the Earth's surface mass load, result in deformation of
the Earth's topography through a series of interconnected processes known as glacial isostatic adjustment (GIA). An
important impact of GIA for the purpose of ice-sheet modelling is the subsidence of the bedrock topography beneath an ice-
sheet. The rate of the solid Earth response towards isostatic equilibrium, which can range from centuries to millions of years,
is viscoelastic in nature as a result of the rheological structure of the Earth and specific pattern of ice loading. In order to
simulate the first-order effects of GIA on bedrock topography, we couple the ice-sheet model to a simple Elastic Lithosphere



Relaxing Asthenosphere (ELRA) model which approximates this response by assuming a fully elastic lithosphere above a
uniformly viscous asthenosphere (Kachuck et al. 2020). A relaxation time of 3000 years is applied in this model based on
previous studies (Pollard and Deconto 2012).

In running FAMOUS-BISICLES, a 10 times acceleration is applied to the ice sheet model to save computational cost
(Gregory et al. 2012, Ziemen et al. 2014). In this method, the ice sheet model is integrated for 10 years using 1-year of
climate simulation by FAMOUS. Gregory et al. (2012) and Gregory et al. (2020) show that 10 times acceleration has a small
to negligible impact on the simulated ice sheet evolution, supporting the use of this technique.
**2.2 Experimental design**
Our experiments mainly follow the protocol of PMIP4 LGM simulations (Kageyama et al. 2017, 2021), which specifies the
insolation, atmospheric concentration of Greenhouse gases ($CO_2$=190 ppm, $CH_4$=375 ppb, $NO_2$=200 ppb, all by volume) and
configurations of continental ice sheets. With respect to ice sheets, in our setup, only the Eurasian and Antarctica ice sheets
are fixed to the reconstruction of GLAC-1D (Tarasove et al. 2012) as the Laurentide and Greenland ice sheets are simulated
with BISICLES. While the protocol specifies the insolation forcing of 21 ka BP, here we use the insolation of 23 ka since the
ice sheet at the LGM is likely still adjusting to earlier forcing (Abe-Ouchi et al. 2013).

For calibrating the slab heat convergence (Section 2.1), we use the SST and sea-ice climatology from a previous LGM
simulation performed with HadCM3, shown in Fig. 1 (Izumi et al., 2022). The simulated SST field exhibits a cold LGM
climate, having a global cooling of 6.5 K. This value is similar to Tierney et al. (2020), who estimate 6.5 K to 5.7 K. For
simplicity of design and clarity of interpretation, the oceanic heat flux is fixed among all the ensemble simulations, thus
assuming no changes in the oceanic heat transport in response to the different parameter values in each member.

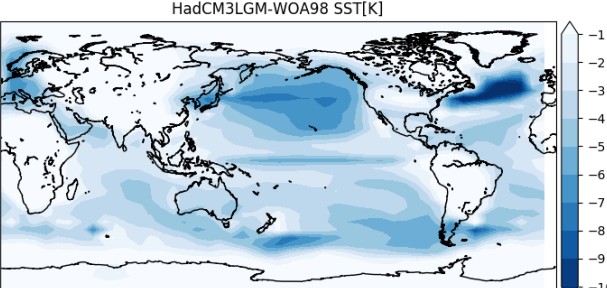


Fig. 1 Annual mean sea surface temperature anomaly fields (K, colour) between a HadCM3 LGM simulation and modern
observation (World Ocean Atlas 1998). The sea surface temperature field from HadCM3 is used as the target sea surface
condition for our prescribed slab ocean setup.

We perform 200-member ensemble simulations by varying16 parameter values associated with climate and ice dynamics, as
summarised in Table 1, using a Latin-hypercube sampling method (Williamson 2015). The choice and the range of the
parameter values in FAMOUS are modified following Gregoire et al. (2012) and Gandy et al. (2023). In BISICLES, the
range of sliding law parameters are modified following sensitivity experiments of Gandy et al. (2019). For *drain*, which
specifies the vertical till-stored drainage rate, the value is very uncertain and hence we varied it to ensure that the till of the
interior of the ice sheet remains dry. Much lower values for *drain*, as used in Gandy et al. (2019) in their simulation of the
much smaller British-Irish ice sheet, result in unphysically wet basal conditions and fast sliding in our simulations so we



used a higher range. For *n*, which specifies the coefficient in Glen's flow law, the range is selected in a practical way;
applying a high value increases the calculation time by more than 10 times due to very large ice velocities and the resulting
refinement in several locations. Hence, the range of *n* is necessarily capped for its upper limit at 3.1, where our technical
tests indicated that the simulations will most likely complete within a feasible run length (two months of wallclock time).
During the ice sheet spin-up phase (see subsection 2.3) we specify a constant SMB. The value of this *smb* is varied across the
ensemble so that the ice volume at the initiation of FAMOUS-BISICLES coupling has a spread of 25 m SLE, which is
similar to the uncertainty in the global ice volume estimates at the LGM (e.g. Abe-Ouchi et al. 2015). Two-hundred sets of
parameter value combinations for these 16 parameters are sampled using a Latin hypercube sampling method, assuming a
uniform value probability across each parameter range, to explore across the full ranges of the 16-dimensional parameter
space.

For simplicity, we apply a single value, spatially uniform basal heat flux of 158 mW/m² and 100W/m² under the grounded
and floating ice respectively. However this needs to be reassessed in the future as both the basal heat flux over the continent
(e.g. Margold et al. 2018) and the ocean can vary spatially.

Table 1 Summary of parameters modified in the ensemble simulations. ND stands for non dimensional.

| Name | Min value | Max value | Unit | Note |
|---|---|---|---|---|
| daice | -0.4 | 0.05 | $K^{-1}$ | Darkening effect of warm surface air temperature on bare ice in the albedo scheme, mimicking water collecting at the surface. Minimum value reduces the bare ice albedo to as low as 0.15 (Smith et al. 2021). |
| fsnow | 350 | 799 | $kg\ m^{-3}$ | Density threshold for snow in the albedo scheme beyond which the surface starts to be regarded as bare ice. Higher values correspond to using brighter albedoes for denser snow and tends to increase ice sheet albedo (Smith et al. 2021). |
| avgr | 0.001 | 0.01 | $\mu m^{-3}$ | Dependence of snow albedo on increasing grain size. Higher value enhances the darkening of snow over time and reduces the snow albedo (Smith et al. 2021). |
| rhcrit | 0.6 | 0.9 | ND | Threshold of relative humidity to form large-scale clouds (Smith, 1990). |
| Vfl | 0.5 | 2.0 | $m\ s^{-1}$ | Speed of ice sedimentation (Heymsfield, 1977). |
| ct | 0.00005 | 0.0004 | $s^{-1}$ | Conversion rate of cloud liquid water droplets to precipitation (Smith, 1990) |
| cw | 0.0001 | 0.002 | $kg\ m^{-3}$ | Threshold value of cloud liquid water for formation of precipitation (Smith, 1990). Only values over land are modified. |
| entcoef | 0.6 | 6.0 | ND | Entrainment rate coefficient. Higher value enhances mixing of an ascending convective plume with ambient dry air. |
| tgrad | -0.01 | -0.002 | $K\ m^{-1}$ | Air temperature lapse rate used during the downscaling to ice sheet surfaces. Larger negative values correspond to stronger lapse rate effects (Smith et al. 2021). |
| alpham | 0.2 | 0.65 | ND | The lowest value of albedo in the sea ice scheme. |
| seaice | 0.00015 | 0.00035 | $m^2\ s^{-1}$ | Efficiency of heat exchange between the base of sea ice and ocean. Higher value increases the heat flux and causes a retreat of sea ice. |





| beta | 20000 | 60000 | $Pa\ m^{-1/3}a^{1/3}$ | Coefficient in Weertman-friction law. Higher value corresponds to stronger friction between the ice base and the dry bedrock (Gandy et al. 2019). |
|---|---|---|---|---|
| coef | 0.4 | 0.6 | $ND$ | Coefficient in Coulomb-friction law (Gandy et al. 2019). |
| drain | 0.2 | 0.6 | $m\ yr^{-1}$ | Magnitude of drainage removing water from the till. Higher value removes water rapidly from the till hence increases the Coulomb-friction (Gandy et al. 2019). |
| smb | 0.01 | 0.1 | $m\ yr^{-1}$ | Magnitude of temporally constant and spatially uniform surface mass balance applied during the standalone BISICLES spin-up. Higher values result in a larger ice sheet at the beginning of the FAMOUS-BISICLES coupled simulation. |
| n | 2.6 | 3.1 | $ND$ | Coefficient in Glen's flow law. |



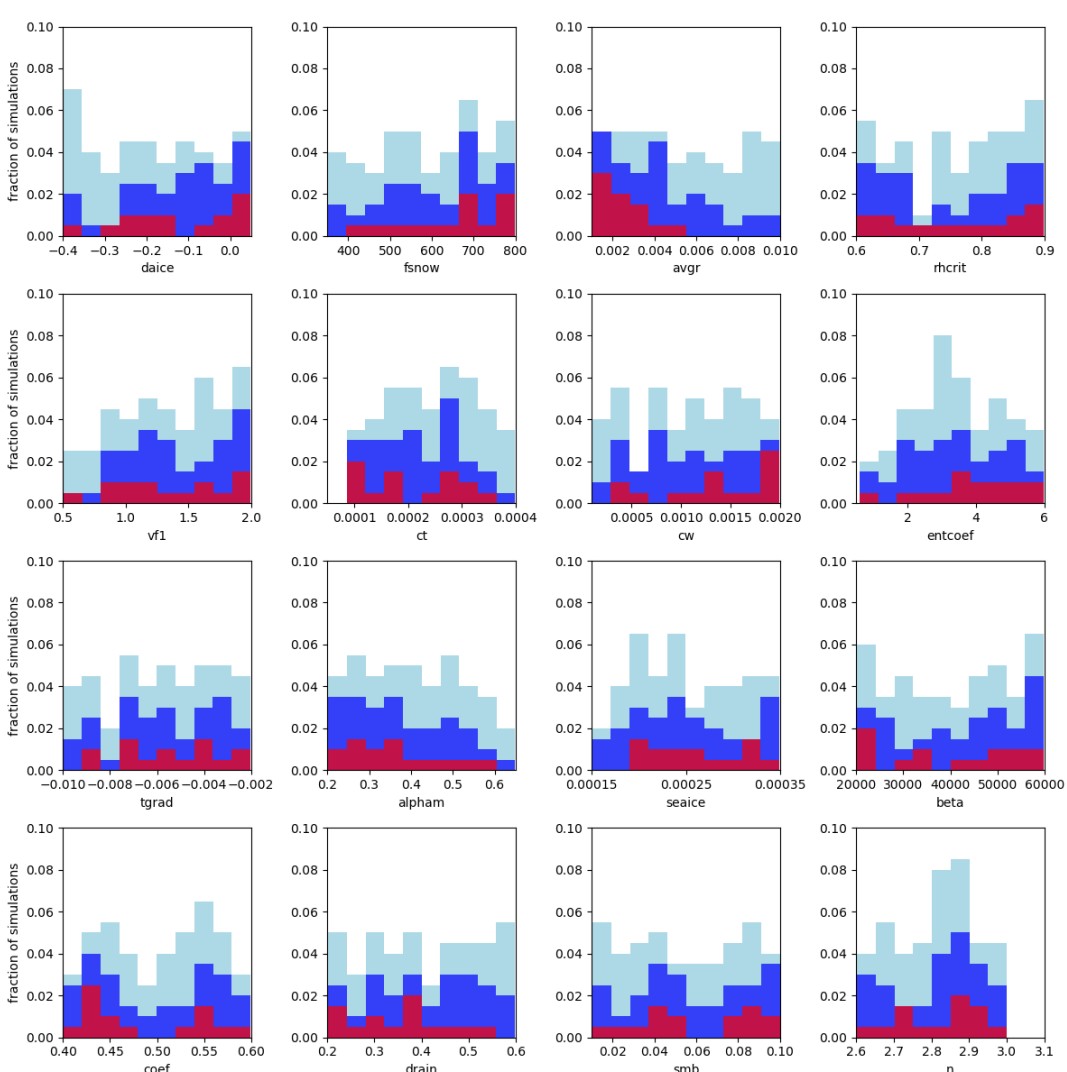




Fig. 2 Fraction of the 200 simulations which satisfy the constraints as a function of each of the parameters. 200 members are
uniformly distributed in each parameter range based on the latin-hypercube sampling method (approximately 20 simulations
per each parameter bin). Light blue: ensemble members satisfying the global temperature constraint, Dark blue: ensemble
members satisfying the global temperature and total ice volume constraints, Red: ensemble members satisfying southern
North American ice sheet margin constraints in addition to global temperature and total ice volume constraints.
**2.3 Integration procedure**
Model simulations are all initiated from a static, isothermal (ice temperature 253 K) ice sheet and bedrock topography of
21ka BP of GLAC-1D (Fig. 3a, f, Tarasov et al. 2012). The simulations have two phases. First, there is an initial 5000 ice
sheet year spin-up with stand-alone BISICLES, where the ice sheet model parameter values are chosen according to the
ensemble Latin Hypercube sampling, but the associated climate parameter values are not used because there is no climate
model. In place of the climate model, a constant-in-time surface mass balance (*smb*, Table 1) and atmospheric surface
temperature of 253K are applied uniformly over the ice. Note that the ice temperature is allowed to evolve in the simulation.
The *smb* value is varied across the ensemble to produce a variety of total ice volumes (Fig. S1), because total ice volume is
highly uncertain in reconstructions and could be important given the dependence of ice sheet simulation on initial conditions
(Abe-Ouchi et al. 2013). The spin-up phase also gives the ice sheet model physics time to adjust from the prescribed initial
condition, i.e. it allows BISICLES to smooth out the blocky surface of the ice sheet reconstruction, providing some stability
to the simulations when they are subsequently coupled to the climate (FAMOUS) in the second phase. By the end of the
spin-up phase, 200 unique ice sheets have been modelled, providing the starting condition for simulations with BISICLES
coupled to FAMOUS in the second phase. In FAMOUS-BISICLES, *smb* is redundant and the climate parameters chosen by
Latin Hypercube are used in FAMOUS, with the same ice sheet parameter combinations as in the spin-up phase. In the
second phase, the simulations run another 5000 ice sheet (500 climate) years, which is insufficient to reach a quasi-
equilibrium state, but sufficiently long to see the effects of important parameters on the climate and ice sheets. For some of
the best performing simulations, the integration is extended for another 5000 ice years. However, the configuration of the ice
sheet showed only modest further changes (Fig. S2).

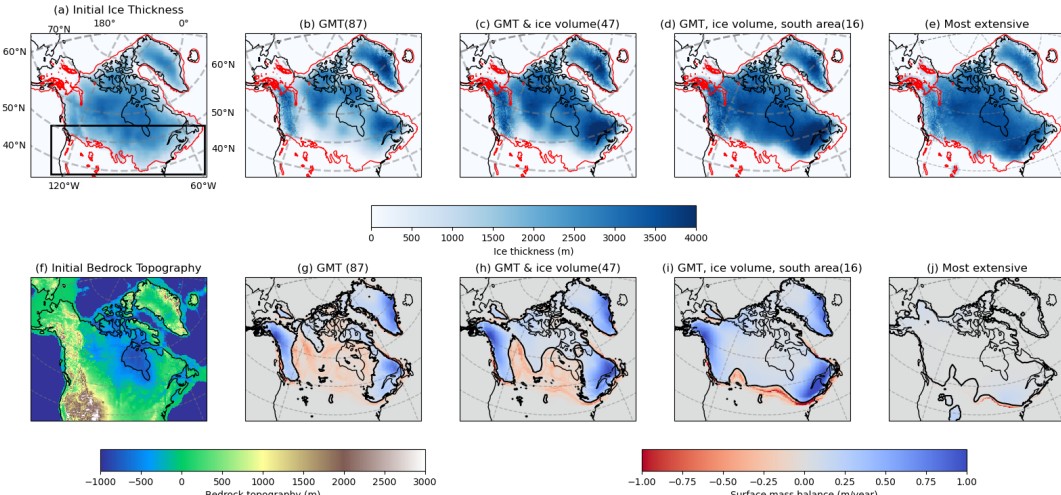

Fig. 3 Spatial maps of the initial condition for the ice sheet model, and results from the FAMOUS-BISICLES ensemble after
5000 ice sheet years. (a) ice topography [m] and (f) bedrock topography [m] from Tarasov et al. (2012). (b-e/top) Spatial
maps of surface altitude [m] and (g-j/bottom) surface mass balance [m/year] from ensemble means. (b, g) 87 members

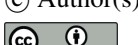



satisfying the global mean temperature constraint, (c, h) 47 members satisfying both global mean temperature and ice
volume constraints, (d, i) 16 members having the largest southern extent of North American ice sheet that satisfies
temperature and volume constraints and (e, j) the member with most extensive southern ice area in the ensemble simulations.
The thin black contour corresponds to the modern coastline, whereas the thick black contour in (g-j) corresponds to the zero
line of SMB. Red contours in (a)-(e) correspond to the ice extent of Dalton et al. (2020). Black rectangle in (a) shows the
region where the southern extent of the North American ice sheet is calculated (e.g. Fig. 11).
**2.4 Constraints**
Three metrics are used to evaluate the large-scale feature of the ensemble simulations. These are the annual mean LGM
global surface air temperature, total ice volume of the North American and Greenland ice sheets and the southern extent of
the North American ice sheet.

For the global temperature, we create our LGM constraint by adding estimates of the LGM global cooling to the
Preindustrial global temperature. The Preindustrial temperature is 13.7 °C (1880-1900, NOAA National Centers for
Environmental Information (2023)) with an uncertainty of $\pm$ 0.1˚C (one standard deviation of global temperature during this
period).  According to previous studies, the LGM global cooling relative to the Preindustrial has a range of -1.7˚C to -8.3˚C
(e.g., -1.7˚C to -3.7˚C with a probability of 90% in Schmittner et al. (2011) and -4.6˚C to -8.3˚C with a probability of 90% in
Holden et al. (2010), see Fig. 4a in Tierney et al. 2020). Assuming the LGM cooling is normally distributed, this gives a
mean cooling of 5 °C $\pm$ 3.3˚C with a probability of 90% (one standard deviation is $\pm$ 2.0˚C). Combining the uncertainties
associated with the Preindustrial global temperature and the LGM global cooling gives one standard deviation of the
uncertainty of
$$\sqrt{(0.1)^2 + (2.0)^2} = \pm\ 2.0\text{˚C}$$

in the actual LGM temperature (66% probability). To be conservative and take into account model uncertainty, we apply
three standard deviations ($\pm$ 6.0˚C) as the uncertainty ranges. This gives an actual LGM temperature of approximately 2.7
˚C to 14.7 ˚C (8.7˚C $\pm$ 6.0˚C), with a probability of  at least 99% (Pukelsheim 1994).

For the total ice volume constraint, previous studies have suggested that the volume of the North American ice sheet was
likely to be higher than 60 m sea level equivalent (c.f. Abe-Ouchi et al. 2015). To account for model uncertainty and to be
conservative, we apply a minimum reasonable North American ice volume of 50 m SLE as a constraint.

The southern extent of the North American ice sheet is used to select the best performing simulations, rather than as a strict
constraint, because all ensemble members show a smaller southern area of the ice sheet than reconstructions (see Section
4.1). Areas of grid cells covered by the ice sheet in the box shown in Fig. 3a are calculated. Simulations with the southern
area covering 60% of the reconstruction (Dalton et al. 2020) are considered to satisfy our constraint.

In the end, sixteen simulations simultaneously satisfy our constraints on temperature, ice volume and extent.
**3. Results**
**3.1 Response of the Global temperature**
Fig. 4 summarises the temporal evolution of annual mean global mean temperature in the ensemble of simulations. After the
first 300 ice sheet years, climates reach a quasi-equilibrium. The results show a wide variety of simulated global



temperatures, ranging from -10˚C to 40˚C. Such a wide range is frequently observed under parameter ensemble simulations
(e.g. Joshi et al. 2010, Gregoire et al. 2011). The diverse response of global temperature is largely explained by two
parameters in the cloud schemes; *ct* in the large-scale condensation scheme and *entcoef* in the cumulus convection scheme
(Fig. 5). The correlation coefficients of these parameters with the global temperature at ice years 200-290 are 0.622 for *ct*
and -0.574 for *entcoef*, respectively. In contrast, other parameters appear to have a smaller effect, according to the correlation
analysis (Fig. 5). For the sea ice albedo, this relatively muted sensitivity may be related to the use of a slab ocean model,
which underestimates the strong interactions between sea ice and oceanic heat transport over the Southern Ocean that
amplifies the surface cooling at high latitudes (Ogura et al. 2004, Zhu et al. 2021). Including a dynamical ocean may increase
the importance of sea ice albedo on the global temperature, as shown by Gregoire et al. (2011).

Roles of *ct* and *entcoef* in governing global temperature are further explored by means of a pair plot in Fig. 6. This figure
compares the relationship of these two parameters to global temperature. The results show a positive correlation between
global-scale warming and *ct*, which is associated with an increase in precipitation efficiency, reducing the life cycle of mid-
latitude clouds, causing a decrease in the cloud cover and a decrease in the planetary albedo. As a result, more shortwave
radiation is absorbed and the planet warms (Joshi et al. 2010, Sherriff-Tadano et al. 2023). Conversely, global-scale
warming occurs with decreasing *entcoef* (Fig. 6), as the entrainment rate of ambient dry air in the tropics reduces, and the
vertical transport of moisture to the high troposphere and lower stratosphere enhances. The planet then warms up due to the
strong greenhouse gas effect of the water vapour (Joshi et al. 2010). Similar responses are observed in Joshi et al. (2010),
who performed ensemble simulations under modern and future climates and showed that low values of *entcoef* were
unrealistic based on the amount of water vapour in the lower stratosphere. Consistently, ensemble members with very low
values of *entcoef* are more likely to be ruled out for producing implausible global mean temperatures, depending on the
effect of the combinations of the other parameters (Fig. 6). For ensemble members satisfying the temperature constraint
(black outlined coloured dots in Fig. 6), the overall cooling and warming effects of *ct* and *entcoef* are largely cancelled out
by each other.



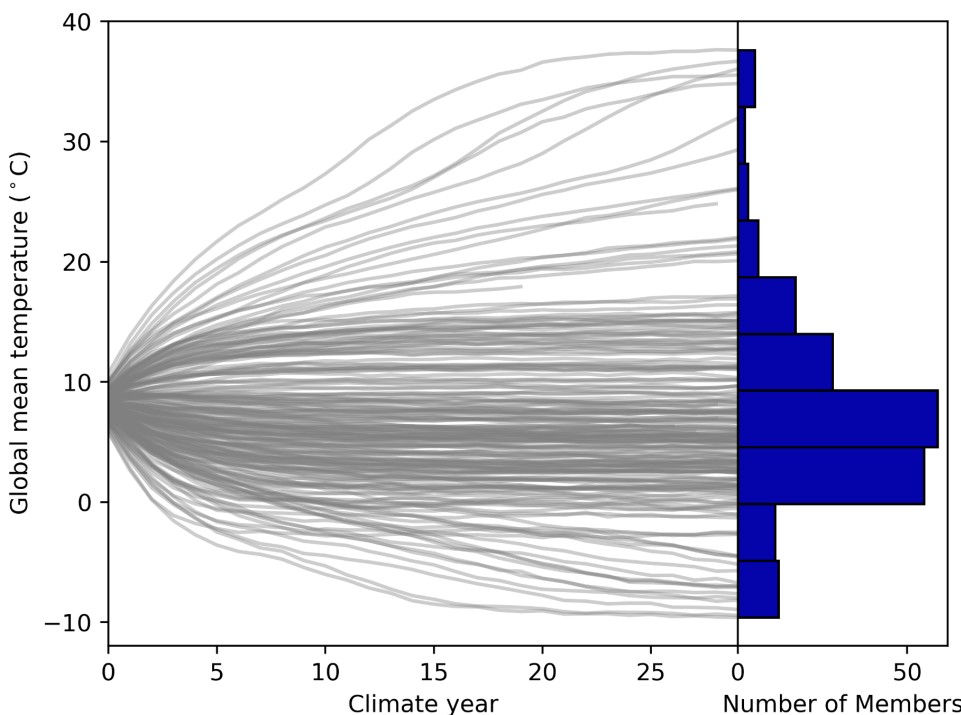

Fig. 4 Evolution of global mean annual temperature in the Famous-Bisicles ensemble of simulations. Each grey line

represents one ensemble member. Results from the first 300 ice years (30 climate years) are shown. Histograms on the right

show the number of simulations in each temperature bin.



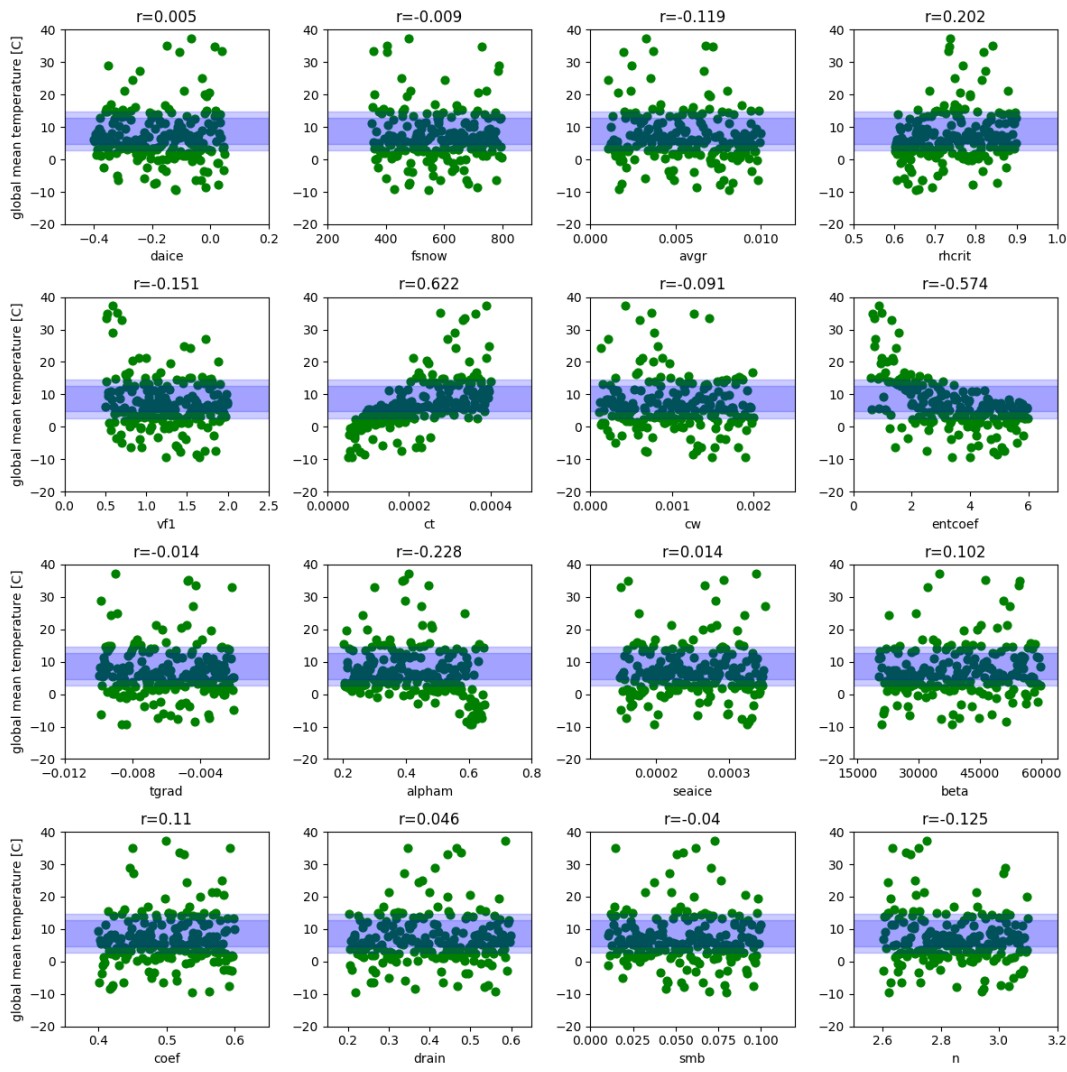

Fig. 5 Relationship between global mean temperature averaged over ice years 200-290 (climate years 20-29) and each

parameter value. Correlation values are displayed above each panel. The uncertainties in global mean annual surface air

temperature is shaded blue (three standard deviations for light blue and two standard deviations for dark blue).





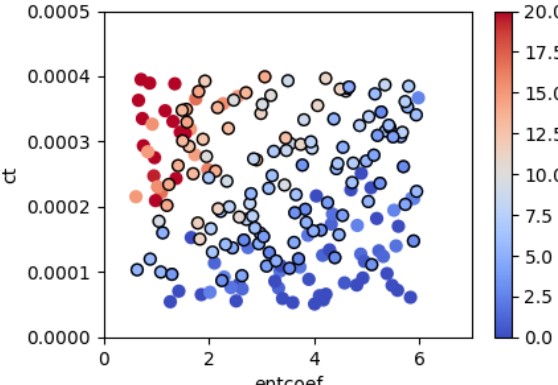


Fig. 6 Pair plot analysis exploring the combined effects of *ct* (precipitation efficiency in the large-scale condensation
scheme) and *entcoef* (entrainment rate in the cumulus convection scheme) on global mean annual surface air temperature
(colours, ˚C). Filled circles outlined in black are those satisfying the temperature evaluation criterion.
**3.2 Response of the North American ice sheet and absolute ice volume**
Similar to the diversity in simulated global mean temperature, the evolution of the ice sheet after the coupling to FAMOUS
shows a wide range of responses (Fig. 7). Starting from absolute ice sheet volumes of 80 to 105 m SLE (sum of North
American and Greenland ice sheets), the ensemble members produce absolute ice volumes between 0 and 120 m SLE at the
end of the 5000-ice year integration. In some simulations, even the Greenland ice sheet disappears completely associated
with the very high global temperature (Fig. 4). Note that some simulations with high *n* values or very warm climates (that
cause all of the ice to rapidly disappear) crash during the integration. In total, 139 members (~ 70% of the ensemble)
complete the entire 5000 ice years. Eighty-seven members satisfy the global temperature constraint (Fig. 5 and Fig. 6) , and
47 members also satisfy the North American ice volume constraint of at least 50 m SLE. The additional constraint on the
southern extent of the North American ice sheet selects the 16 best performing simulations (Fig. 2).



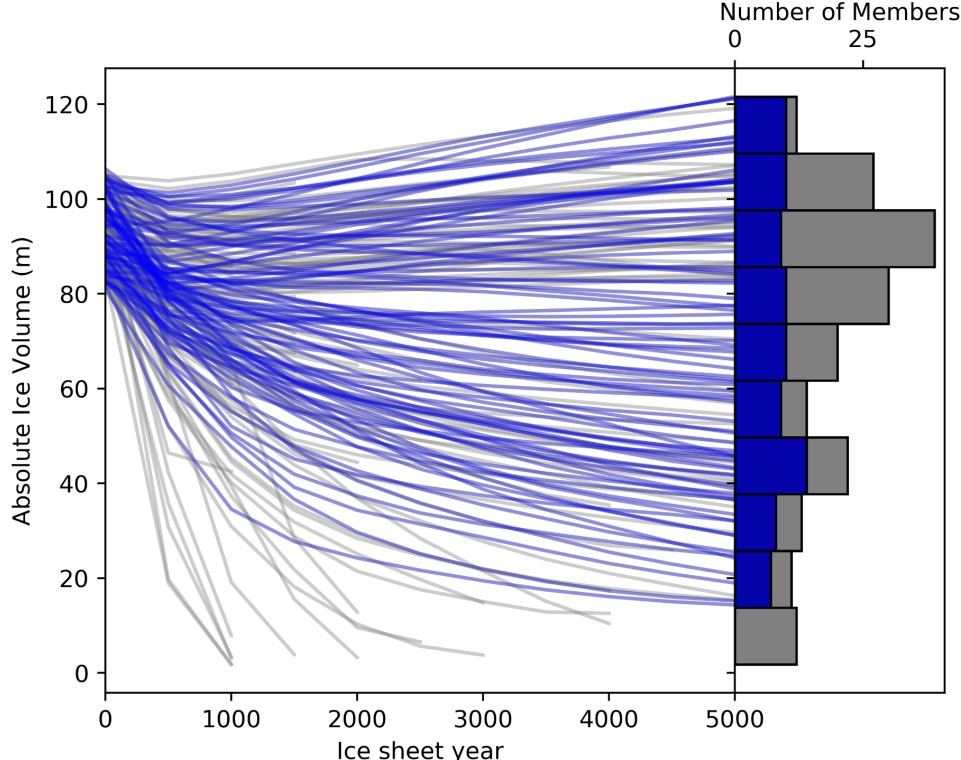

Fig.7 Evolution of absolute ice volume of North America and Greenland in the FAMOUS-BISICLES LGM ensemble. Note that the ice volume of the entire Greenland is included. Each grey line represents one ensemble member. Blue lines are the members satisfying our chosen global temperature evaluation criteria. Histograms on the right show the number of simulations in each temperature bin; grey: all members and blue: members satisfying the global temperature constraint.

To explore which parameters are causing the variety of outcomes for the simulated North American ice volume, scatter plot and correlation analyses are performed (Fig. 8). Here, the ensemble members that both satisfy the global temperature constraint and have completed 5000 ice years are used (87 members). The analysis shows important impacts from parameters in our ice sheet surface albedo scheme that have a direct influence on the albedo that is diagnosed for bare ice or uncompacted snow surfaces; *avgr* (snow ageing effect), *daice* (melt pond effect), and *fsnow* (the weighting of snow and ice albedo based on the density of snow) showing correlations of -0.56, -0.475 and 0.372, respectively, with ice volume (see Table 1 for the effects of each parameter). Similar results are obtained for the analysis on the southern extent of the North American ice sheet (Fig. S4).

Additional analysis exploring the combined effect of these three parameters reveals a strong dependence between *daice* and *fsnow* (Fig. S7); the ice volume is less sensitive to *daice* when *fsnow* has a large value. This is reasonable as a large value of *fsnow* means that most of the snow/ice will be diagnosed as snow due to the high value of density threshold. As a result, the darkening effect for the old ice (*daice*) has only a minor influence.

The effects of other climate parameters are weaker compared to those of albedo parameters. Among these, *ct* shows the largest correlation value of -0.325. This is reasonable since the low value of *ct* corresponds to a colder global climate (Fig.



5), hence a colder local climate over the ice sheet, allowing the large ice sheet to be sustained (see also section 3.4 and Fig.
11). On the other hand, the 87 not-ruled-out-yet simulations are relatively insensitive to *entcoef* (Fig. 8). This may in part be
due to the screening out effect of ensemble members with low values of *entcoef* that causes drastically warm climates. We
should also note that the cloud parameters exert some local influences on accumulation patterns, e.g. over the Gulf stream
region (Fig. S6); larger values of *ct* and *cw* correspond to an increase in the amount of snowfall in this area. However the
overall low correlation values between *cw* and the ice volume of North America shows a relatively weak effect of
accumulation on the simulated ice volume.

Correlation analysis shows a very weak effect from basal drag parameters (*beta* and *coef*) on the ice volume (Fig. 8) and the
southern extent (Fig. S4). The correlation value of *smb*, which controls the initial ice volume when the coupled climate-ice
sheet phase of each simulation starts, is also low (r=0.22). This suggests only a weak connection between final ice sheet
volume at 5000 years and its initial volume at the beginning of the coupled simulations. This is due to the large
modifications in snow/ice albedo in our ensemble design, which is capable of drastically altering the magnitude of absorbed
solar radiation over the ice sheet (e.g. Abe-Ouchi et al. 2013). For other dynamical ice sheet parameters (*drain* and *n*), the
correlations are generally even lower. Overall, the North American ice sheet volume is much less sensitive to uncertainty in
ice sheet dynamics than ice sheet albedo and climate in our parameter space.

Interestingly, we find that the main results showing the importance of albedo parameters can be found in the first 500 ice
sheet years by analysing the relation of ice volume changes and each parameter (106 members, Fig. S3). Similar results are
also obtained by Gregory et al. (2020), who show that the SMB of the first 100 years can be a good predictor of the final
steady state ice sheet mass of modern and future Greenland. These results suggest that significant computational cost could
be saved for at least an initial exploration of model sensitivity to uncertain parameter values (e.g. if designing a multi-wave
ensemble experiment).





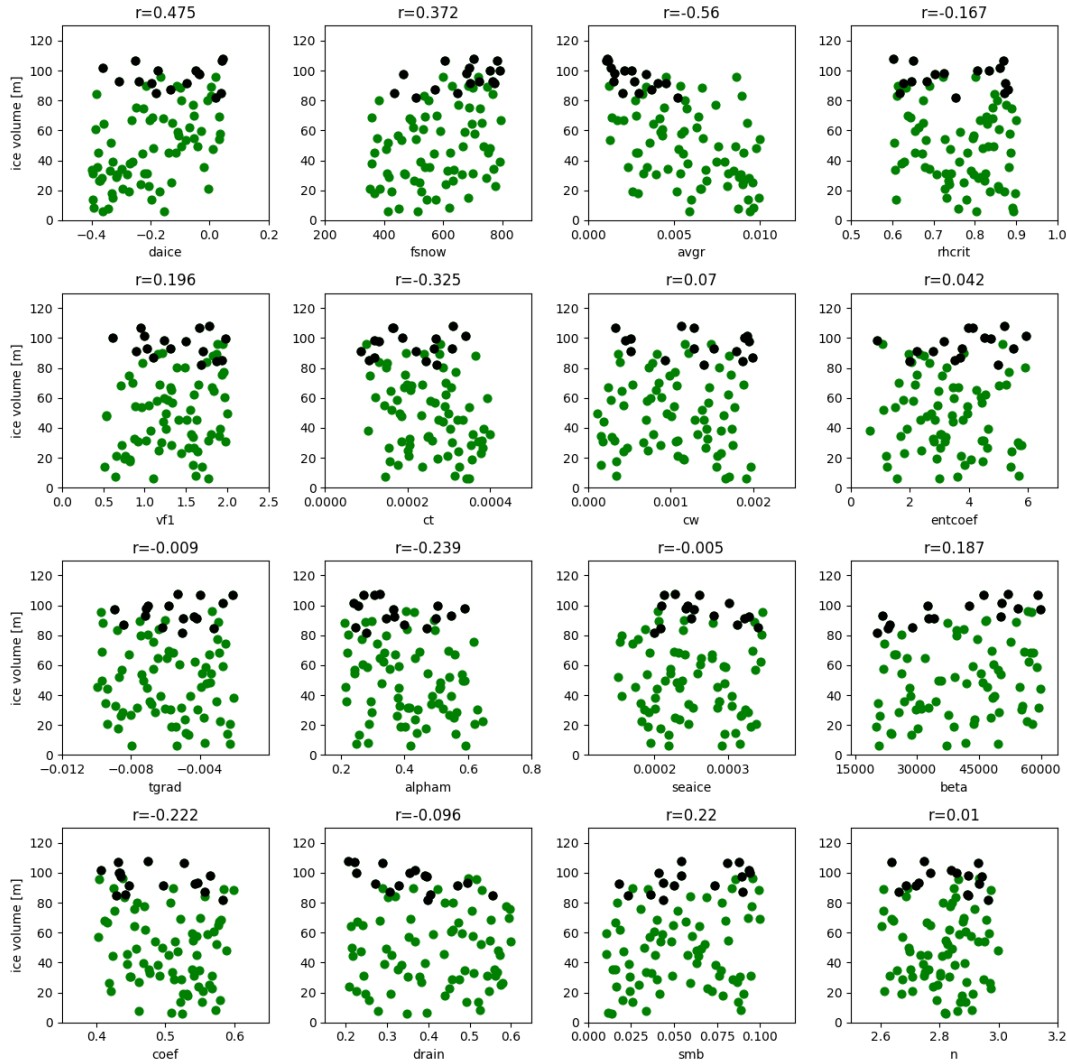

Fig. 8 Relationship between North American ice volume at 5000 ice years in FAMOUS-BISICLES and each perturbed

parameter. Only those ensemble members that satisfy the global temperature constraint are used. Correlation values are

displayed above each panel. Black dots correspond to the best sixteen members.

To explore our preferred parameter space that produces good climate and ice sheets at the LGM, the distributions of

parameters satisfying the applied constraints are examined (Fig. 2). Results show that some of the parameter ranges may be

ruled out due to poor resulting simulation performance, such as values below 400 of *fsnow*, values above 0.006 of *avgr*,

values below 0.00008 of *ct* and values above 3.0 of *n*. Additionally, from Fig. S7, a combination of low values in both *daice*

and *fsnow* may be ruled out. Runs that satisfy the constraints tend to have parameters that lead to higher albedo values. For

other parameters,  it is shown that values across any individual parameter range in the ensemble can produce reasonable

global temperatures and ice sheets, depending on their combination with others.

The performance of the simulated ice extent in the best sixteen simulations (Fig. 3d) is further evaluated against the ice

extent reconstruction from Dalton et al. (2020, red contour in Fig. 3d). In general, the average of the best sixteen simulations



reproduces the overall ice extent of the North American ice sheet reasonably well; e.g. performances over the northern
margin and the southern margin west of 110˚W and east of 80˚W are reasonable. Also the performance is much better
compared to means of members that satisfies the global temperature and the ice volume constraints (Fig. 3b, c). In contrast,
the main differences between the best sixteen simulations and the reconstruction appear over the southern margin at 110˚W -
80˚W, where the model underestimates the area of the ice sheet. Another difference can be found over Alaska, where the
model overestimates the ice sheet area and thickness (Fig. 3d). These features are commonly observed in ice sheet model
simulations coupled to a low-resolution atmospheric model and will be discussed in section 4.1.

Away from the southern margin, the best performing FAMOUS-BISICLES simulations tend to lack sufficient ice at the
eastern margin, where an ice shelf should exist (Fig. 3d). This is associated with the strong and uniform basal ice shelf
melting applied in this study. The basal melting around the coastal area largely depends on the configuration of the
continental shelf as well as the ambient ocean temperature, as shown by studies on the Antarctic ice sheet (e.g. Obase et al.
2017). Future work could undertake additional sensitivity experiments changing the magnitudes and patterns of the basal
melting to further explore this point.
**3.3 Responses of the Greenland ice sheet**
The Greenland ice sheet also shows various responses to modifications in the parameters in the ensemble of simulations,
ranging from 8 m SLE to 15 m SLE (Fig. 9). The simulated  range is similar to the range in the reconstructions suggesting
9.3 m to 12.3 m SLE (7.3 m + 2~5 m SLE, Clark and Mix 2002, Lecavalier et al. 2014, Bradley et al. 2018, Tabone et al.
2018), while the model overestimates the higher band.

Interestingly, the results show a different sensitivity to the model parameters we vary compared to the North American ice
sheet (Fig. 9). The variations in the ice volume are mostly explained by changes in *beta*, where higher values increase the
friction between the ice sheet and the bedrock at a cold ice base. This acts to increase the ice volume by reducing the amount
of ice transported to its margin which then calves at the continental shelf, and hence by inducing thickening of the ice sheet
interior.

The lower sensitivity of the Greenland ice sheet to albedo parameters comes from different climatic conditions compared to
North America. In North America, the large area is covered by negative surface mass balance (Fig. 3g) as the summer
temperature can be close to freezing point in the simulations (Fig. 10). Hence, albedo parameters cause a drastic difference
since they control the magnitude of the negative SMB over North America (Fig. 8). In contrast, the Greenland ice sheet is
covered by colder conditions in summer (Fig. 10), hence most surface areas have positive surface mass balance (Fig. 9).
Under this condition, the amount of  the ice loss is determined by the amount of ice transported from the interior to its edge,
which then calves. As a result, the ice volume is mainly driven by *beta* since it controls the transport of ice under the cold ice
base.

Previous studies have shown that basal melting of ice shelves by the underlying ocean is also important in controlling
Greenland ice sheet volume at the LGM in their coupled ice shelf-ice sheet models (Bradley et al. 2018, Tabone et al. 2018).
In this study, however, a constant value was given for the ice shelf basal melting. Conducting ensemble simulations with
variations in the amount of ice shelf melting may enable us to explore the relative importance.



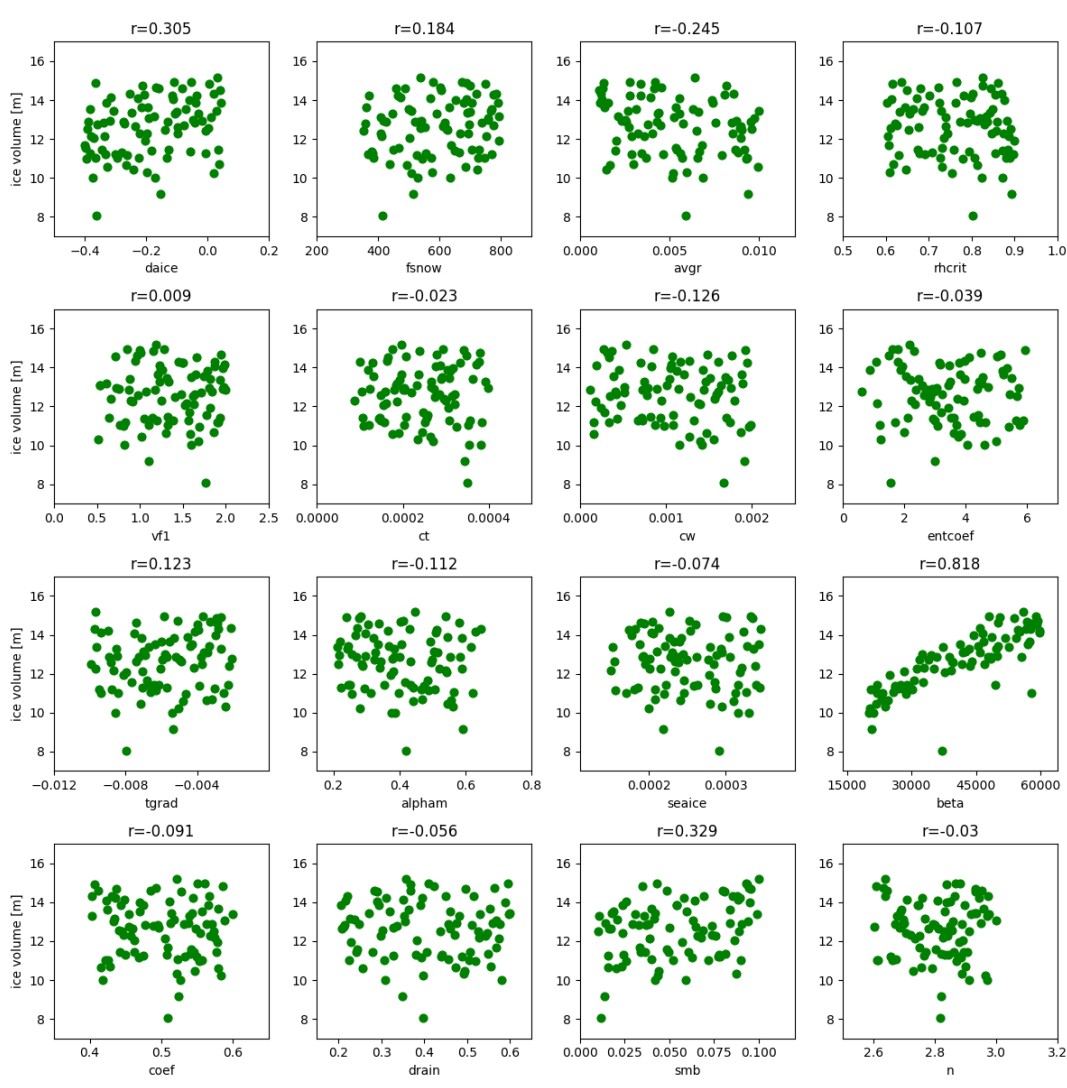


Fig.9 Relation of ice volume of Greenland at 5000 ice years in FAMOUS-BISICLES and each parameter. Ensemble

members satisfying the global temperature constraint are used. Correlation values are displayed on the top of each panel.

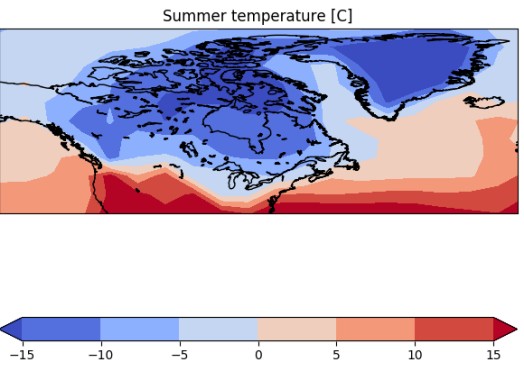




Fig.10 Summer surface air temperature [˚C] over North America and Greenland, averaged over all ensemble members
satisfying the global temperature constraint.
**3.4 Effects of global mean temperature on ice sheet volume**
The sensitivity of the ice sheets to the reasonable LGM global temperature range (2.7˚C-14.7˚C) is explored to see any
possible relationship between them (Fig. 11). The results show a high correlation between the global temperature and North
American ice volume/southern extent; colder climates correspond to larger and more extensive ice sheets (Fig. 11a, b). This
is not a surprise since a large uncertainty of ± 6.0˚C is applied to the global temperature. Reducing the uncertainty level to
two sigma (8.7˚C ± 4.0˚C, black dots in Fig. 11) weakens the correlation between the global temperature and the North
American ice volume/southern extent to -0.193 and -0.285, respectively. Nevertheless, the correlation analysis still shows
some sensitivity of the southern extent of the North American ice sheet to global temperature (Fig. 11b), where a colder
global climate tends to produce a more extensive ice sheet in the south. In other words, it can also be said that it is hard to get
an extensive southern North American ice sheet under warm LGM global temperature (above 12.0˚C), irrespective of the
albedo parameters, which demonstrates the value of constraining the upper band of real LGM temperatures for simulating
the North American ice sheet well.

The Greenland ice sheet appears to be insensitive to the reasonable LGM global temperature range (2.7˚C-14.7˚C), which is
consistent with the dominant role of basal sliding in controlling the ice volume. Reducing the uncertainty level to two sigma
(8.7˚C ± 4.0˚C, black dots in Fig. 11) increases the correlation value to 0.259 possibly associated with an increase in snow
fall following the warming climate, however the effect is much weaker compared to the effect of basal sliding.

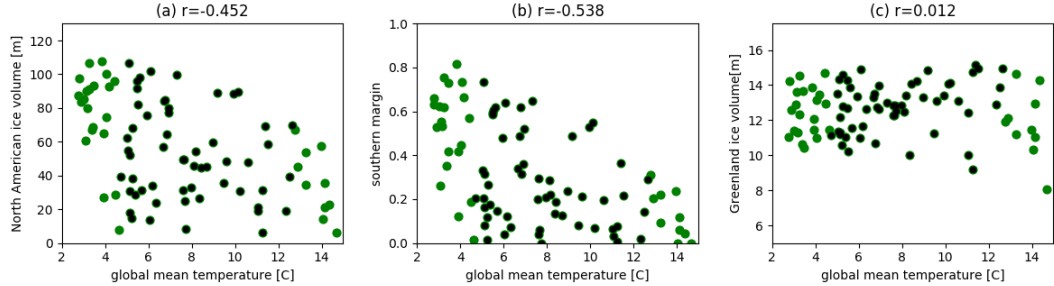


Fig. 11 Relationship between global mean annual surface air temperature [˚C] and Ice sheet variables. (a) North American
ice sheet volume [m], (b) Ratio of southern extent of the North American ice sheet compared to Dalton et al. (2020) and (c)
Greenland ice sheet volume [m]. Ensemble members that satisfy the global temperature constraint and have run 5000 ice
sheet years are used (87 members). Correlation values are also shown in each figure. Black dots show results within the two
sigma uncertainty in the LGM global temperature (8.7˚C ± 4.0˚C).
**4. Discussion**
**4.1 How could FAMOUS-BISICLES be made to reproduce the southern extent of the North American ice sheet?**
A recent study by Gandy et al. (2023) performed a similar ensemble simulation with FAMOUS-GLIMMER with fixed SSTs
instead of FAMOUS-BISICLES coupled to a slab ocean model used in this study. Our findings here are consistent with them
in that the ice extent is sensitive to choices of parameters in the snow and ice albedo scheme and that both models
underestimate the southern extent of the North American ice sheet, especially the so-called 'lobe' characteristics. To
investigate the possibility of the model being able to reproduce the full extent of the southern margin of the North American



ice sheet, we analyse in detail the ensemble member that has the most extensive southern margin, disregarding our imposed climate plausibility constraints (Fig. 3e). In the simulation, the performance of the southern extent of the North American ice sheet improves and becomes closer to the reconstructed area due to the very cold climate, whose absolute global temperature is -7.4℃. Yet even in this very cold simulation, the model cannot maintain the 'lobe' characteristics of the North American ice extent as far south as the reconstructions.

So, how might we reproduce the southern margin of the North American ice sheet? There are several possibilities:

- Finer horizontal resolution in the climate model: during the simulations, FAMOUS-BISICLES loses the thin ice sheet at the south margin abruptly in the first 1000 ice sheet years due to the very large negative SMB simulated in the atmospheric model (e.g. Fig. 13b). As discussed above, applying a high-resolution atmospheric model might be better able to sustain a more southerly ice margin through a stronger stationary wave effect that cools the area (Abe-Ouchi et al. 2007).
- Representation of clouds: Gregory et al. (2012) pointed out the importance of changes in cloud cover over the southern margin of the North American ice sheet on its SMB during the glacial inception. Having a larger cloud cover at the southern margin may help to maintain the ice sheet by reducing the very large negative SMB, while a careful analysis on the physical plausibility needs to be done.
- Improvements in the downscaling scheme: including the effect of strongly stratified boundary layer on the surface temperature during the downscaling may allow a colder surface temperature over ice, which can help sustain the ice sheet at its margin. Incorporation of downscaling of accumulation in FAMOUS-BISICLES can increase the snow fall at the southern margin, which increases the SMB and surface albedo and may help to sustain the ice sheet at the southern margin (e.g. Yamagishi and Abe-Ouchi 2005).
- Higher initial surface elevation: the simulation could be started with a higher initial surface elevation which can be obtained by giving a thicker ice or a higher bedrock topography at the southern margin, allowing for lower surface temperatures due to the higher elevation, although this may not be physically plausible.
- Palaeo-vegetation: the choice of vegetation type for the unglaciated region near to the ice sheet may be relevant. The modern vegetation distribution used in this study may tend to give a warmer condition in this area, unlike tundra, which grows under cold climates and causes a surface cooling (O'ishi and Abe-Ouchi 2013).
- Bedrock conditions: creating a slippery bedrock condition would enhance ice flow from the ice sheet interior towards the margin, and so may be instrumental in redistributing ice outwards.
- Longer integration of the model: extending the integration of FAMOUS-BISICLES may help to redistribute the thick ice in the interior to the southern margin. In fact, some of the members, which have been extended for additional 5000 years show a southward expansion (Fig. S2).

It is also possible that the concept of the southern margin being in a quasi-equilibrium state with the LGM forcing may not be valid, and that it may be reflecting several transient ice advancing events that occurred during the recent glacial period (and preceding the LGM)(e.g. Pico et al. 2017, Gowan et al. 2021, Bradley et al. 2023). We speculate that such earlier southward ice advance may allow a more expansive southern ice sheet to establish, before rebalancing with the insolation forcing. In this case, running a long transient simulation, rather than performing equilibrium-type LGM simulations, may be essential for achieving the target southern margin extent.

**4.2 Performances of ice streams**

The positions of our simulated ice streams in the best sixteen ensemble members are evaluated against the reconstruction by Margold et al. (2018) (Fig. 12 and Fig. S5). The figure depicts that BISICLES shows regions of relatively high ice velocities (or ice streams) at various sites, despite the relatively low resolution of the model (16 km at finest grid) and the relatively



short integration period. Specifically, most members reproduce high ice velocities at the margin over the Baffin Bay area. In
addition, the simulation of ice streams facing the Arctic Ocean is encouraging (Fig. 12, S5). However, once again the
southern margin is tricky to get right, and our ice stream behaviour there is somewhat diffuse, not picking up the
characteristic 'lobe' structure of the reconstructions (Margold et al. 2018). Over the Eastern North American ice sheet, the
model captures some large glaciers such as Laurentian Channel (25), Placentia Bay-Halibut Channel (133) and Hopedale
Saddle (168), while none of the best sixteen ensemble members simulate the large ice stream that flows to the Labrador Sea
from the present-day Hudson Bay area. These poorly represented ice stream features may be caused by low resolution of the
smallest ice sheet refinement (16 km, e.g. Gandy et al. 2019), too-short integration and misrepresentation of the surface type
of till (Gowan et al. 2019). With the last point, the amount of till water calculated prognostically in the simulations appears
small, hence most areas use the Weertman sliding law. An increase in the basal melting, a choice of a smaller value for
*drain* or incorporating a spatially variable Weertman coefficient map based on geological evidence may help to improve the
performance of the ice streams.  Nevertheless, the model does show some reasonable potential in simulating North American
ice streams considering the relatively low resolution as well as the explicit calculation of basal drag.

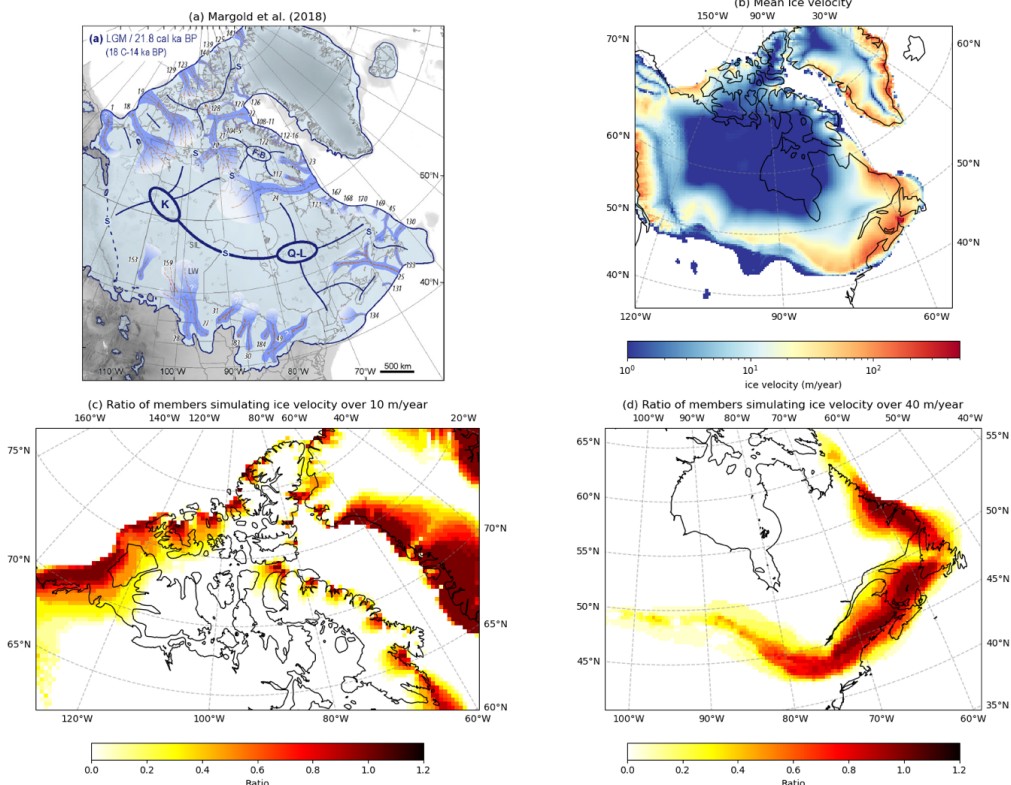

Fig.12 Comparison of ice velocity [m/year, colour] between (a) Reconstruction (Margold et al. 2018, adapted from Fig. 5 of
Margold et al. 2018) and (b) the mean of best sixteen members. (c) and (d) show the ratio of numbers of members simulating
ice velocity beyond 10m/year for (c) and 40m/year for (d), respectively. Ratio of 1.0 means all the sixteen members simulate
ice velocities of those values.
**4.3 Effects of biases in the simulated climate**



Some of the simulations in the ensemble exhibit a local melting of the ice sheet from parts of the interior outwards, which is
unusual, as ice sheets usually melt from their margins, where the surface temperature is close to the freezing point (e.g. Figs
3c and 13). This phenomenon is caused by biases in the atmospheric model, which are then amplified by the downscaling
method and a positive feedback from the coupling. First, the model has a warm summer temperature bias over the ice sheet
interior. As a result, large parts of the central North American ice sheet have a temperature above -10 ˚C despite the surface
elevation exceeding 2000 m (Fig. 10). A similar feature was pointed out by Smith et al. (2021) using the same model under
the modern Greenland ice sheet, which produced a higher ELA (around 2 km high) compared to a high resolution regional
atmospheric model (at about 1 km high). Second, because the downscaling of SMB strongly depends on the elevation, a
local change in surface elevation can induce a local negative surface mass balance if the surface temperature calculated in the
FAMOUS grid points are close to the freezing point. This example is shown in Fig. 13, where a negative SMB can be found
at the local minimums of surface elevation, despite the elevation exceeding 2000 m. The initial negative SMB then kicks in a
strong positive feedback where melting of snow reduces the albedo and results in more energy absorption. As a result, the ice
elevation starts to decrease and causes additional positive feedback similar to saddle node collapse (Gregoire et al. 2012).
This way of downscaling the climate model output works well for modern Greenland, especially at low elevation where the
SMB has very strong elevation dependence. However, at the higher altitudes achieved by the LGM North American ice
sheet, SMB may be more greatly  affected by other factors such as wind speeds, as suggested by studies on Antarctica (Van
Liefferinge et al. 2021). Hence, further improvements in the downscaling method at higher elevation could help to reduce the
impact of the climate biases.

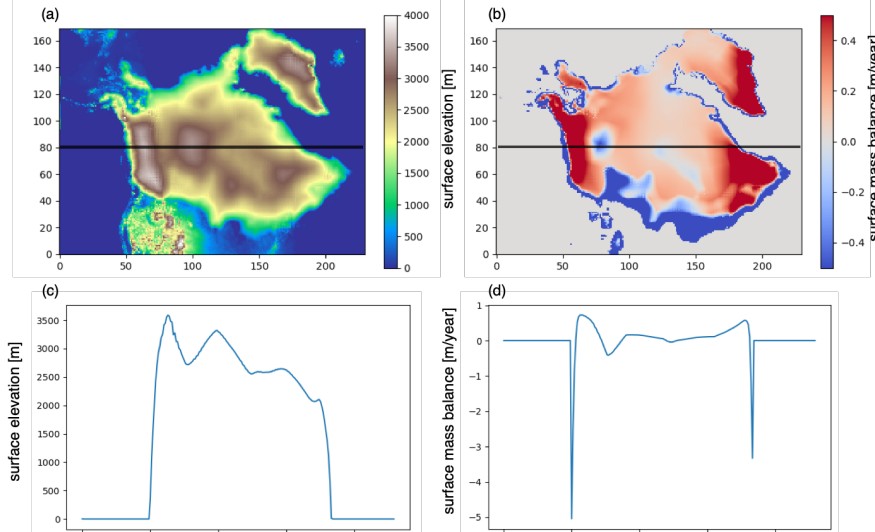

Fig.13 An example of local ice melting in the interior of the ice sheet. (a) Surface topography [m] and (b) Surface mass
balance [m/year]. Height zonal cross-section of (c) surface topography and (d) surface mass balance at y=80 are shown.
**5. Conclusion**
In this paper we have presented a large ensemble of simulations of the North American and Greenland ice sheets and climate
of the LGM, performed with a coupled atmosphere-ice sheet-slab ocean model FAMOUS-BISICLES, a version of the
FAMOUS-ice model developed by Smith et al. (2021). The experiment consists of a 200-member perturbed parameter
ensemble, where the values of 16 parameters associated with climate and ice dynamics were varied using a Latin-hypercube
sampling method. The simulated results are evaluated against the LGM global mean surface air temperature, the North



American ice volume and the southern extent of the North American ice sheet. In the ensemble, the global temperature is
controlled by a combination of precipitation efficiency in the large-scale condensation and entrainment rate in the cumulus
convection, consistent with previous FAMOUS simulations of modern climate (Joshi et al. 2010). Under reasonable LGM
global temperature conditions, we find that the surface albedo exerts the strongest control on North American ice volume. In
contrast, the ice volume of Greenland is found to be mainly controlled by the Weertman coefficient in the basal sliding law.
The different sensitivity of these ice sheets to the model's physical parameter values mainly comes from different climatic
conditions; the North American ice sheet being generally warmer hence has a larger area of negative SMB, which is affected
by the albedo. In contrast, most parts of the Greenland ice sheet are covered by a very cold atmosphere, hence the ice sheet
volume is more affected by the calving at its margin, the total amount of which is controlled by the magnitude of the basal
sliding law that affects the amount of ice transported to the margins. These differences between the North American and
Greenland ice sheets provide an important take-home message on model performance, suggesting that for best flexibility
(i.e., the ability to simulate conditions largely different from today), simulators should be calibrated under substantially
different climate and ice sheet conditions and tested out-of-sample.

Analysis of the relationship between the North American southern ice extent and global temperature with the uncertainty
level of two sigma (8.7˚C $\pm$ 4.0˚C) shows a slightly weak relation. Nevertheless, we find that it is hard to get an extensive
southern North American ice sheet under warm LGM global temperature (above 12.0˚C), irrespective of the albedo
parameters in our model. This demonstrates the value of constraining the upper band of real LGM temperatures for
simulating the North American ice sheet well.

Based on our plausibility constraints, the model produces sixteen 'acceptable' simulations with reasonable global
temperature and North American ice sheet. These simulations show the most extensive southern margin under reasonable
LGM temperature and ice volume, but, as with many LGM ice sheet simulations, are not sufficiently expansive at the
southern margin and overestimate ice volume in Alaska. Both of these features are likely attributable to the underestimation
of the stationary wave effect (Roe and Lindzen 2001, Abe-Ouchi et al. 2007), and may be improved upon/overcome by
increasing the climate model resolution. We find that the model cannot reproduce the southern margin of the ice sheet over
the 5000-year simulation even if the absolute global temperature is as cold as -7.4˚C, and it is also possible that more
accurate representation of the palaeo vegetation, different treatments of ice sheet sliding and downscaling method of the
SMB, or a different spin-up procedure could improve the simulated southerly ice sheet extent.

Our results show that warm summer temperature biases in the interior of the ice sheet as well as the downscaling method of
SMB based on elevation can cause strong local melting of the ice sheet from the interior outwards. More complex treatment
of the atmospheric conditions and surface mass balance in the ice sheet interior could improve this, and may be especially
important when applying the model to the Antarctic ice sheet.

Lastly, the strong sensitivities of the North American ice sheet to albedo at the LGM may imply a potential constraint on the
future Greenland ice sheet by constraining the formulation and behaviour of albedo schemes for climate and ice sheet models
under relatively warm climates. Running similar ensemble simulations for the modern and future Greenland ice sheet will
provide an important data set to directly connect the simulations of past climates and ice sheets to those of the modern and
future. Using such data, we may be able to explore whether simulations of past climate-ice sheet conditions can more tightly
constrain or increase the confidence of projection of future sea level rise .
**Code and data availability**



The simulation data of FAMOUS-BISICLES used in this study will be available in a public database.
**Author contribution**
Sam Sherriff-Tadano (Data curation, Formal Analysis, Investigation, Methodology, Validation, Visualization, Writing-
original draft), Ruza Ivanovic (Conceptualisation, Funding Acquisition, Investigation, Methodology, Project Administration,
Resources, Software, Supervision, Writing - review and editing), Lauren Gregoire (Conceptualisation, Funding Acquisition,
Investigation, Methodology, Project Administration, Resources, Software, Supervision, Writing - review and editing),
Charlotte Lang (Data curation, Formal Analysis, Investigation, Methodology, Writing-review and editing), Niall Gandy
(Data curation, Formal Analysis, Investigation, Methodology, Writing-review and editing), Tamsin Edwards (Funding
acquisition, Methodology, Writing – original draft). Robin S. Smith (Conceptualisation, Funding Acquisition, Methodology,
Project Administration, Resources, Software, Supervision, Writing - review and editing), Jonathan Gregory
(conceptualization, funding acquisition, methodology, software, writing - review and editing), Oliver Pollard (Methodology,
Visualization, Writing - review and editing)
**Competing interest**
The authors declare no competing interests.
**Acknowledgements**
This work was undertaken on ARC4, part of the High Performance Computing Facilities at the University of Leeds,
ARCHER2 and JASMIN. RFI, RSS, JG, TE, CL and SST were funded by "RiSICMAP", NERC Standard Grant
NE/T007443/1. NG, LJG, RFI were funded by "SMB-Gen" UKRI Future Leaders Fellowship MR/S016961/1. We are also
grateful to Richard Rigby for his assistance in setting up the simulation. SST also thanks Ayako Abe-Ouchi, Miren Vizcaino,
Heiko Goelzer and Jonathan Owen for constructive discussion.

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
