# Peer review of "Large ensemble simulations of the North American and Greenland ice"

_EGUsphere, 2023_

## Referee Comment (RC1)

**Review of: Large ensemble simulations of the North American and Greenland ice sheets at the Last Glacial Maximum with a coupled atmospheric general circulation-ice sheet model by Sherriff-Tadano *et al.**

Evan J. Gowan

evangowan@gmail.com

Sherrif-Tadano *et al.* present the results of a coupled ice sheet-ocean-atmosphere model (FAMOUS-BISICLES) applied to the Last Glacial Maximum (LGM) in North America and Europe. Their goal is to see what parameters control the extent and volume of the North American and Greenland ice sheets, using an ensemble of 16 parameters, related to ice sheet sliding and flow, albedo, and surface climate. The 16 parameters are randomly varied using Latin Hypercube sampling to produce an ensemble of 200 simulations. The simulations are first run for a 5000 year spinup starting with the GLAC-1D reconstruction using constant surface mass balance and air temperature using only the BISICLES ice sheet model. After the spinup, the model is coupled using FAMOUS and run for another 5000 years, after which the results are compared. One of the results of these simulations is that the global temperatures are strongly controlled by the parameters $ct$ and $entcoef$, both related to the parameterization of clouds. Of the 200 simulations, 87 satisfy the set threshold for global temperature after 5000 model years. Overall, the resulting ice volume for North America is weakly dependent on the varied parameters, with only a small correlation with parameters related to albedo ($avgr$, $daice$, $fsnow$). In contrast, the Greenland Ice Sheet shows a very strong relationship with the ice-ground friction parameters, $\beta$. Global temperature provides a control on the southern extent of the North American ice sheets, while Greenland is relatively insensitive.

I appreciate the advance this study makes to create a coupled ice sheet-climate-ocean model that can be applied to paleo-simulations. Being able to explore a wide range of factors to discern ice sheet behavior is an exciting development. The main weakness of this study is with the use of Latin Hypercube sampling to determine the values of the parameters of the model simulations. I have mentioned this in a previous review of this model (Gandy *et al.*, 2021) that by varying a large number of parameters simultaneously, it becomes difficult to discern the relative impact that each parameter has on the evolution of the the the simulation. This is the case here (*i.e.* Figures 8 and 9), where aside from the sliding parameter $\beta$ for the Greenland Ice Sheet, there is only a weak relationship between the varied parameters and resulting ice sheet volume. Part of this is because some parameters (*e.g.* the parameters related to albedo and sliding) can cancel each other out. It would have been easier to determine the relationship between variables if a smaller number were selected, then varied in a controlled way. I suppose this may not have been known at the start of the study that this kind of cancellation would happen. However, I think a change of study design would lead to a more interesting result. I think the current results should be published, though I hope the authors consider this in the future. At the very least, the results from Greenland, where the ice sheet volume is controlled by basal conditions rather than global climate, is a very interesting result.

I think one way to improve this study would be to break up the North American ice sheets into smaller regions and see if different sectors are sensitive to specific parameters. For instance, I would

expect the Cordilleran Ice Sheet, which is underlain by mountainous topography, will be sensitive to $\beta$, similar to the Greenland Ice Sheet. I would also expect that there will be different sensitivities to the parameters for the southern, land terminating part of the Laurentide Ice Sheet, versus the marine terminating eastern part. Similarly, I would expect different sensitivities between the southern Laurentide and the Innuitian/Northern Laurentide Ice Sheets in terms of climatic parameters. Perhaps cluster analysis could also be applied to see if better relationships between the overarching parameters (*e.g.* related to sliding, albedo) can be deduced.

**Southern extent of the Laurentide Ice Sheet and ice streams**

Much of section 4 discusses how the model is unable to reproduce the ice streams and ice lobes that existed in the southern Laurentide Ice Sheet. However, the explanations given ignore what I would consider the most likely reason the ice streams and lobes existed – the presence of ice marginal proglacial lakes (*e.g.* Cutler *et al.*, 2001; Quiquet *et al.*, 2021). The proglacial lakes destabilized the ice sheet and encouraged the flow of ice in much the same way as marine terminating ice streams. The presence of shallow lakes that were insufficient to act as destructive calving margins would have increased the subglacial water pressure, encouraging a decoupling of the ice-bed interface, causing the ice sheet to advance in a lobe. When we added proglacial lakes in the PISM ice sheet model (Hinck *et al.*, 2022), we demonstrated the presence of lakes greatly enhanced ice flow, and we also had some limited success in simulating ice lobe formation in shallow lakes.

If this is correct, then it is not surprising that the FAMOUS-BISICLES model is unable to simulate the southern margin of the Laurentide Ice Sheet or terrestrial ice streams, since it lacks this mechanism. The simulation is also of an LGM climate. Since the ice streams and lobes are largely acknowledged to be a result of ice sheet dynamics rather than climatic impacts (Jennings, 2006), perhaps this should not be a target metric for the success of the model. The extreme southern limit of the ice sheet was not achieved at the LGM, because the dynamics requires large amounts of meltwater, which was inhibited by the cold temperatures at that time. Perhaps a simpler target, such as an ice margin near the Canada-US border, would be better.

**Minor comments**

- Figure 3: please explicitly define "GMT" in the caption.

- Figure 13: I would recommend adding details of which simulation was used to produce this (i.e. what were the atmospheric conditions in this model simulation).

- Some of the references mentioned in the text are not in the reference list, please check.

Best Regards,

Evan J. Gowan

**References**

Cutler, P.M., Mickelson, D.M., Colgan, P.M., MacAyeal, D.R., Parizek, B.R. 2001. Influence of the Great Lakes on the dynamics of the southern Laurentide ice sheet: Numerical experiments. Geology **29**, 1039–1042. https://doi.org/10.1130/0091-7613(2001)029<1039:IOTGLO>2.0.CO;2.

Gandy, N., Gregoire, L.J., Ely, J.C., Cornford, S.L., Clark, C.D., Hodgson, D.M. 2021. Collapse of the Last Eurasian Ice Sheet in the North Sea Modulated by Combined Processes of Ice Flow, Surface Melt, and Marine Ice Sheet Instabilities. Journal of Geophysical Research: Earth Surface **126**, e2020JF005755. https://doi.org/10.1029/2020JF005755.

Hinck, S., Gowan, E.J., Zhang, X., Lohmann, G. 2022. PISM-LakeCC: Implementing an adaptive proglacial lake boundary in an ice sheet model. The Cryosphere **16**, 941–965. https://doi.org/10.5194/tc-16-941-2022.

Jennings, C.E. 2006. Terrestrial ice streams–a view from the lobe. Geomorphology **75**, 100–124. https://doi.org/10.1016/j.geomorph.2005.05.016.

Quiquet, A., Dumas, C., Paillard, D., Ramstein, G., Ritz, C., Roche, D.M. 2021. Deglacial ice sheet instabilities induced by proglacial lakes. Geophysical Research Letters **48**, e2020GL092141. https://doi.org/10.1029/2020GL092141.

---

## Referee Comment (RC2)

Review of manuscript:

*Large ensemble simulations of the North American and Greenland ice sheets at the Last Glacial Maximum with a coupled atmospheric general circulation-ice sheet model*

The authors have presented a comprehensive paper tackling an ongoing issue in climate-ice sheet modelling which is the uncertainty in parameter space within the model. Focusing on the ice sheet southern margin extent, which as the authors state is an ongoing problem to achieve was a very original approach. I enjoyed reading the paper but have several main points that I would like the authors to provide more information for or make small changes in the manuscript.

**Main points:**

**Model evaluation metrics:**

**LGM temperature.** I found the paragraph describing the uncertainty calculation hard to comprehend. There has been several specific publications which have estimate LGM global temperature with an uncertainty range (Tierney, 2o022, Osman). What was the authors reasoning for this approach?

**Ice sheet volume:**

**Lower limit**:50m  The reference the authors have chosen to define their LGM NAIS and GrIS ice sheet volume is old. A lower limit, which the authors use of 60m is from ICE4G, which has been preceded by ICE6G and ICE7G. Both these latter two studies have a larger total ice volume, ~ 76m. Therefore, I do not think a lower limit of 50m is a good value to use. Tarasov et al., 2012 Table1, has published a study exploring a range of LGM NAIS volumes., but there are others. I am not sure how much this lower limit influences the authors parameter space, as from Figure 8, the minimum volume of the 16 parameters > 80m.

**Upper limit:** Have the author considered using an 'upper limit' for the total ice sheet volume. From the Figure 8; some of the best 16 members, (black dots) have total volumes of ~ 110m. Given that this number does not include the Antarctic ice sheet (~10m) or the Eurasian ice sheet (~24m; number from the authors paper), this would produce a total global sea level at this time would be too large, 144m. (rough calculations). This may reduce the possible parameter space, but it will also rule out ice sheet volume that do not appear viable.

**Southern margin extent**
Including the southern margin as a metric to evaluate the ice sheet-climate simulation is an original approach. The extent of the box the authors have used (Figure 2), from my understanding consider a margin that has retreated up to Hudson Bay as reasonable?

**Parameter testing procedure**
The authors have taken the temperature as the primary criteria and then adding ice volume and southern margin extent. I am interested to know if the authors started with a 'ice

volume' if this would have impacted on their results? As this is to some extent, a study focused on the ice sheets.

**Spin up procedure.**
What was the authors reasoning for ice-sheet spin up and then adding in the climate parameters? I understand that running the climate model is computationally expensive, however from the SOM figure including the climate parameters seemed to feedback onto the ice sheet?

**Comments about figures:**
**Figure 11**: I really liked this figure to try and understand how the different criteria used in the study relate. Is this all 200 ensemble members
What I find interesting, which I hope the authors can comment on is in panel (a) the same ice sheet volume, ~ 70m is produced for a GMST between 5C and 12C. Has the ice sheet not thicken? Changed in extent? I am trying to understand the 3 factors together. In terms o the southern margin, about ~ 11 C the southern margin has undergone a large retreat.  Perhaps if the authors plot North American volume vs ice sheet margin this will become apparent.

**Figure S3:**
This is an interesting figure and from my understanding this is after the spin-up procedure (ice sheet only parameters)? If this is the case, the ice volume can reduce by up to 40m? Given than in the ice sheet-only stage (Fig S1) the volume in some simulation increases by ~ 20m, does this climate influence (feedbacks?) reduce this? This possible relates to my above question about spin-up, why not spin up with the climate feedbacks?

**Figures changes:**
The figures with multiple panels are small for the reader to see. This might be the typesetting of the manuscript but can the authors try to increase.

**Figure1:** I would suggest changing the title to a more general phrase.
I am confused how there are SST across the land region? Is it SAT?

**Figure2:**
Can you add a key onto the figure to state: light blue = GMST; dark blue ...

**Figure 3**:
Can you highlight the edge of the actual simulated ice sheet? It is hard to identify where the edge of the ice sheet is (panelsb,c,d,e) without guessing in reference to the ablation area. Does this figure only show grounded ice?

**Figures 4 and 7**:

For these graphs can you add on the limit of GMST and ice volume as you have, for example on Figure 5.

**Minor comments**
Line 217: Laurentide> this is one ice sheet which makes up the LGM North American ice sheet: change to North American

**Terminology:** Can the author clarify from the beginning the difference between FAMOUS-ICE (is this with always an ice sheet? Or just the climate component): FAMOUS-Ice (Gandy et al., 2023) - this is when it is coupled to Glimmer, and FAMOUS-ICE, which then is referred to in the abstract as FAMOUS-BISICLES.
**GMT** - this is a very common abbreviation for other things: please change to GMST, Sat or something else.

---

## Author Comment (AC1)

**Response to Reviewer#1 (Dr Evan Gowan)**

We are grateful to Dr Evan Gowan for all the constructive comments and time for reviewing our manuscript. As described below, we will take all the suggestions by the reviewer into account in the revised manuscript. We also performed additional analysis to address the reviewer's concern. Below, our responses are shown in blue and the comments by the reviewer are shown in black.

Responses to comments:

I appreciate the advance this study makes to create a coupled ice sheet-climate-ocean model that can be applied to paleo-simulations. Being able to explore a wide range of factors to discern ice sheet behavior is an exciting development.

Thank you!

The main weakness of this study is with the use of Latin Hypercube sampling to determine the values of the parameters of the model simulations. I have mentioned this in a previous review of this model (Gandy et al., 2021) that by varying a large number of parameters simultaneously, it becomes difficult to discern the relative impact that each parameter has on the evolution of the the simulation. This is the case here (i.e. Figures 8 and 9), where aside from the sliding parameter β for the Greenland Ice Sheet, there is only a weak relationship between the varied parameters and resulting ice sheet volume. Part of this is because some parameters (e.g. the parameters related to albedo and sliding) can cancel each other out. It would have been easier to determine the relationship between variables if a smaller number were selected, then varied in a controlled way. I suppose this may not have been known at the start of the study that this kind of cancellation would happen. However, I think a change of study design would lead to a more interesting result. I think the current results should be published, though I hope the authors consider this in the future. At the very least, the results from Greenland, where the ice sheet volume is controlled by basal conditions rather than global climate, is a very interesting result.

Thank you for the comment. As the reviewer rightly points out, in such a complex model, it is difficult to tease out the sensitivity of the results to individual model parameters. This is because of the many interactions between the different climate and ice sheet processes in the model, which leads to what the reviewer calls the "cancellation" of the effects to the parameters. We will clarify the reason of the choice of Latin Hypercube sampling in the Method as follows;

"We perform 200-member ensemble simulations by varying16 parameter values associated with climate and ice dynamics, as summarised in Table 1, using a Latin-hypercube sampling method (Williamson 2015). Latin-hypercube sampling technique is useful as it allows us to explore all the uncertain parameter spaces in an efficient way. While some cancellations among parameters can cause lower correlation values between inputs and outputs, the method also provides quantitative insights on the interactions among different parameters (e.g. Fig. 6 and Fig. S7 in this study)."

As the reviewer suggests, performing sensitivity experiments modifying small numbers of parameters in a controlled way are definitely a good way to understand how each parameter affects and interacts with the coupled climate-ice sheet system. Perhaps, combining the Latin-hypercube sampling and the controlled way sampling might be an ideal way, e.g. finding out important

parameters in wave1 with Latin-hypercube sampling and then performing controlled sampling in wave2 or wave3 with smaller sets of parameters.

Another way of doing this is could be to perform a Sobol sensitivity analysis on an ensemble of simulations (Sobol', I. M.: On Sensitivity Estimation for Nonlinear Mathematical Models, Matematicheskoe mod- elirovanie, 2, 112–118, 1990) as we have recently done with an ice sheet and sea level model (Pollard et al., submitted to Quaternary science review)." We will consider doing them in the future!

I think one way to improve this study would be to break up the North American ice sheets into smaller regions and see if different sectors are sensitive to specific parameters. For instance, I would expect the Cordilleran Ice Sheet, which is underlain by mountainous topography, will be sensitive to β, similar to the Greenland Ice Sheet. I would also expect that there will be different sensitivities to the parameters for the southern, land terminating part of the Laurentide Ice Sheet, versus the marine terminating eastern part. Similarly, I would expect different sensitivities between the southern Laurentide and the Innuitian/Northern Laurentide Ice Sheets in terms of climatic parameters. Perhaps cluster analysis could also be applied to see if better relationships between the overarching parameters (e.g. related to sliding, albedo) can be deduced.

This is a very good point! We conducted additional analysis separating the North American ice sheet into seven different sectors (NW, SW, N, M, MS, NE, E in Fig. R1). Table R1 summaries the relation among parameters and ice volumes at each sector. While the most important parameters remained to be the albedo ones (*daice* and *avgr*), we found that *beta* has an increased influence over SW and M, as suggested by the reviewer. We will add a following subsection in the revised manuscript.

"3.5 Localities in the effect of parameters

The different sensitivities to parameters between the North American and Greenland ice sheets imply that similar variations in sensitivity to parameters may exist between different local regions within the huge North American ice sheet. To explore this point, we separate the North American ice sheet into seven different sectors (NW, SW, N, M, MS, NE, E), where a substantial amount of ice remains in the ensemble mean of members satisfying the GMST constraint (Fig. 12). Results are summarized in Table 2. While the albedo parameters remain the most important ones (*daice* and *avgr*) in each region, we find that *beta* has an increased influence in SW and M. These areas either exhibit a mountainous bedrock topography or have very thick ice, hence can be more affected by the basal sliding parameters. Additionally, we find that *ct* has a relatively strong influence on the northern (N) and eastern (E) parts of the North American ice sheet. Our analysis indicates some variation in regional sensitivities to climate and ice sheet parameters in different sectors of the ice sheet sectors. Further analysis beyond the scope of this study would be required to explore this regional dependency in detail."

[Figure]

Fig. R1 Six different areas (NW, SW, N, M, NE and E) of the North American ice sheet used for the additional analysis (black rectangle). Blue shades show the mean ice thickness [m, colour] of members satisfying the global mean surface temperature constraint.

Table R1 Four most influential parameters on ice volumes at different regions. Values in the bracket show the correlation. For the Southern Extent, results from Fig. S4 are used.

| Region | 1 | 2 | 3 | 4 |
|---|---|---|---|---|
| NW | *avgr* (-0.48) | *fsnow* (0.47) | *daice* (0.4) | *ct* (-0.25) |
| SW | *fsnow* (0.42) | *daice* (0.4) | *beta* (0.39) | *avgr* (-0.35) |
| N | *avgr* (-0.44) | *daice* (0.37) | *ct* (-0.36) | *fsnow* (0.28) |
| M | *daice* (0.53) | *avgr* (-0.49) | *beta (0.29)* | *ct* (-0.25) |
| MS | *avgr* (-0.58) | *daice* (0.47) | *fsnow* (0.39) | *ct* (-0.30) |
| NE | *avgr* (-0.52) | *daice* (0.49) | *smb* (0.30) | *fsnow* (0.26) |
| E | *avgr* (-0.48) | *daice* (0.43) | *fsnow* (0.33) | *ct* (-0.30) |
| Southern Extent | *avgr* (-0.52) | *daice* (0.41) | *fsnow* (0.36) | *ct* (-0.33) |

Southern extent of the Laurentide Ice Sheet and ice streams

Much of section 4 discusses how the model is unable to reproduce the ice streams and ice lobes that existed in the southern Laurentide Ice Sheet. However, the explanations given ignore what I would

consider the most likely reason the ice streams and lobes existed – the presence of ice marginal proglacial lakes (e.g. Cutler et al., 2001; Quiquet et al., 2021). The proglacial lakes destabilized the ice sheet and encouraged the flow of ice in much the same way as marine terminating ice streams. The presence of shallow lakes that were insufficient to act as destructive calving margins would have increased the subglacial water pressure, encouraging a decoupling of the ice-bed interface, causing the ice sheet to advance in a lobe. When we added proglacial lakes in the PISM ice sheet model (Hinck et al., 2022), we demonstrated the presence of lakes greatly enhanced ice flow, and we also had some limited success in simulating ice lobe formation in shallow lakes.

If this is correct, then it is not surprising that the FAMOUS-BISICLES model is unable to simulate the southern margin of the Laurentide Ice Sheet or terrestrial ice streams, since it lacks this mechanism. The simulation is also of an LGM climate. Since the ice streams and lobes are largely acknowledged to be a result of ice sheet dynamics rather than climatic impacts (Jennings, 2006), perhaps this should not be a target metric for the success of the model. The extreme southern limit of the ice sheet was not achieved at the LGM, because the dynamics requires large amounts of meltwater, which was inhibited by the cold temperatures at that time. Perhaps a simpler target, such as an ice margin near the Canada-US border, would be better.

Thanks for the comment! We will add the following sentence in Discussion 4.1.

"Bedrock conditions: creating a slippery bedrock condition would enhance ice flow from the ice sheet interior towards the margin, and so may be instrumental in redistributing ice outwards. In this regard, adding a scheme that allows the generation of proglacial lakes and increase ice flow at the southern margin would help advance the lobe (Hinck et al. 2022)."

While the ice dynamics part is essential in completing the ice lobe, we do think that the climate part is also important for simulating the lobe. This is because, without simulating appropriate climate, there won't be any ice close to the lobe in the first place. In this regard, we think that understanding the relation of climate-albedo parameters and the southern extent of the North American ice sheet is meaningful and important.

For the last point, we conducted analysis focusing on the performance of the ice volume near the Canada-US border (MS in Fig. R1). It turns out that the members showing extensive southern margin in Fig. S4 are the members simulating the largest ice volume at the MS region. Hence, we will keep using the same box in Fig. 3 as the metric for the southern margin.

• Figure 3: please explicitly define "GMT" in the caption.

Done!

• Figure 13: I would recommend adding details of which simulation was used to produce this (i.e. what were the atmospheric conditions in this model simulation).

Done!

• Some of the references mentioned in the text are not in the reference list, please check.

Done! Thanks!

---

## Author Comment (AC2)

**Response to Reviewer#2 (Dr Sarah Bradley)**

We are grateful to Dr Sarah Bradley for the constructive comments and the time for reviewing our manuscript. As described below, we will take all the suggestions by the reviewer into account in the revised manuscript. We also performed additional analysis to address the reviewer's concern. Our responses will be shown in blue and the comments by the reviewer will be shown in black.

Responses to comments:
The authors have presented a comprehensive paper tackling an ongoing issue in climate-ice sheet modelling which is the uncertainty in parameter space within the model. Focusing on the ice sheet southern margin extent, which as the authors state is an ongoing problem to achieve was a very original approach. I enjoyed reading the paper but have several main points that I would like the authors to provide more information for or make small changes in the manuscript.

Thank you!

**LGM temperature.** I found the paragraph describing the uncertainty calculation hard to comprehend. There has been several specific publications which have estimate LGM global temperature with an uncertainty range (Tierney, 2022, Osman). What was the authors reasoning for this approach?

The estimated magnitude of LGM cooling differs among studies and it has a range from 1.7°C to 8.3°C cooling as summarized in Tierney et al. (2020). In this study, we wanted to cover all the possibilities of the actual LGM temperature for the temperature constraint. Therefore, we decided to take all the previous studies into account in an objective way, including Tierney et al. (2020), rather than being subjective and picking one particular study. This caused a wide range of acceptable LGM actual temperature in this study. We will add the following sentence in the revised manuscript;

"According to previous studies, the LGM global cooling relative to the Preindustrial has a range of -1.7°C to -8.3°C (e.g., -1.7°C to -3.7°C with a probability of 90% in Schmittner et al. (2011) and -4.6 °C to -8.3°C with a probability of 90% in Holden et al. (2010), see Fig. 4a in Tierney et al. 2020). To objectively cover all the possibilities, we take into account all of these studies to define our range of plausible LGM GMST."

**Lower limit**:50m The reference the authors have chosen to define their LGM NAIS and GrIS ice sheet volume is old. A lower limit, which the authors use of 60m is from ICE4G, which has been preceded by ICE6G and ICE7G. Both these latter two studies have a larger total ice volume, ~ 76m. Therefore, I do not think a lower limit of 50m is a good value to use. Tarasov et al., 2012 Table1, has published a study exploring a range of LGM NAIS volumes., but there are others. I am not sure how much this lower limit influences the authors parameter space, as from Figure 8, the minimum volume of the 16 parameters > 80m.

First of all, we apologize to the reviewer that the explanation of ice volume constraint was unclear in the original manuscript. While we used the North American ice volume as the constraint, some of the sentences described that the constraint was on the North American and Greenland ice volumes. We will clarify in the revised manuscript that the ice volume constraint is only applied to the North American ice sheet.

Second of all, as the reviewer pointed out, the lower ice volume limit didn't have an effect on the selection of best performing members due to the stronger southern extent constraint. In fact, we performed a test analysis changing the value from 50 to 60, but did not find any major changes.

Nevertheless, we will change the lower ice volume limit of the North American ice sheet to 60 m following the reviewer's advice. Accordingly, we will update Figs. 2 and 3 using 60m SLE as the minimum ice volume constraint for the North American ice sheet.

**Upper limit:** Have the author considered using an 'upper limit' for the total ice sheet volume. From the Figure 8; some of the best 16 members, (black dots) have total volumes of ~ 110m. Given that this number does not include the Antarctic ice sheet (~10m) or the Eurasian ice sheet (~24m; number from the authors paper), this would produce a total global sea level at this time would be too large, 144m. (rough calculations). This may reduce the possible parameter space, but it will also rule out ice sheet volume that do not appear viable.

We reanalyzed Figs. 2 and 8 with the max ice volume limit of 100m SLE. This caused a reduction of numbers of best 16 members to 10, however the preferred parameter space did not change.

In general, equilibrium LGM simulations tend to overestimate the ice volume once the simulation has a net positive SMB since it keeps growing during the integration (e.g. Alder and Hostetler 2019). In this regard, setting an upper limit can be tricky. Therefore, we would like to add the following sentence in the revised manuscript to inform the readers that future study should consider this point.

"Applying an upper ice volume limit may also be important in constraining the parameter space. However, in general, equilibrium LGM simulations tend to overestimate the ice volume if once the simulation has a net positive SMB (e.g. Alder and Hostetler 2019). In this regard, setting an upper limit can be tricky, and therefore needs to be examined in a different experimental set-up."

**Southern margin extent:** Including the southern margin as a metric to evaluate the ice sheet-climate simulation is an original approach. The extent of the box the authors have used (Figure 2), from my understanding consider a margin that has retreated up to Hudson Bay as reasonable?

We did not have an intention to say that members showing ice sheet beyond Hudson Bay are reasonable, but now that the reviewer has mentioned, it might be an valid point to mention. We will add the following sentence in section 2.4;

"This area corresponds to the south of the Hudson Bay".

**Parameter testing procedure:** The authors have taken the temperature as the primary criteria and then adding ice volume and southern margin extent. I am interested to know if the authors started with a 'ice volume' if this would have impacted on their results? As this is to some extent, a study focused on the ice sheets.

Thanks for the comment. We will add the following subsection in the revised manuscript.

"**3.6 Sensitivity of influential parameters to individual constraints**

Applying our three simulation constraints simultaneously may be hiding relationships that exist between model parameters and simulation behaviour. We perform additional analyses to explore how each constraint individually affects the relationship between our model parameters and North American ice sheet volume. In the case of no-constraints (139 members), the albedo parameters are important, but the influence from $ct$ becomes more important (Table 3). This is due to the increased range of GMST allowed by varying $ct$ (Fig. 5). Having a much colder or warmer climate allows the ice sheets to grow or melt, and the resulting feedback further enhances the role of $ct$. In contrast, most members with extremely warm climates crashed during the 5000 year simulation. This means

that, *entcoef* does not appear to have so large an effect on ice sheet volume directly, unlike its importance in setting the GMST.

In the case of applying only the ice sheet volume constraint (73 members), *avgr* and *fsnow* still show relatively high correlations with ice sheet volume. However their influence is less than when GMST constraint alone is applied (Table 3). The ice volume constraint alone results in a preferred selection of members exhibiting colder climates (46 members have a GMST below 4 ˚C). As a result, the members are less sensitive to albedo related parameters.

When the southern extent constraint alone is applied, 33 members remain. Similar to above, members satisfying this condition tend to have very cold climates, where 24 members have GMST colder than 4˚C and 14 members colder than 0.63˚C. In this case, *avgr* and *beta* appear to be most influential. This may imply that snow albedo and basal conditions play an important role in maintaining an extensive ice sheet once the climate allows the ice sheet to reach this size. Further discussion on the maintenance of the southern margin of the North American ice sheet is in subsection 4.1.“

Table R2 Effects of constraints on the relation of parameters and North American ice sheet volume at year 5000. The four most influential parameters on ice volumes are shown.

| Region | 1 | 2 | 3 | 4 |
|---|---|---|---|---|
| No Constraint (139 members) | *daice* (0.51) | *avgr* (-0.45) | *ct* (0.45) | *fsnow* (0.35) |
| GMST-alone (87 members) | *avgr* (-0.56) | *daice* (0.48) | *fsnow* (0.37) | *ct* (-0.33) |
| Min Ice volume-alone (73 members) | *avgr* (-0.39) | *fsnow* (0.33) | *smb* (0.33) | *daice* (0.24) |
| Southern Extent-alone (33 members) | *avgr* (-0.71) | *beta* (0.51) | *smb* (0.44) | *fsnow* (0.39) |

**Spin up procedure:** What was the authors reasoning for ice-sheet spin up and then adding in the climate parameters? I understand that running the climate model is computationally expensive, however from the SOM figure including the climate parameters seemed to feedback onto the ice sheet?

We have two reasons for this. The first one is related to the efficiency and the initial stability of the simulation. The geometry of GLAC1D ice sheet used as the initial condition can have some areas with blocky and cliffy surfaces. As a result, running the first couple of hundred years can take a while since BISICLES adjust dx, dy and dt depending on the ice velocity and others. In this situation, coupling with the climate model made the entire simulation extremely long and starting with the BISICLES-only simulation was much more efficient (L271-273). Secondly, we wanted to introduce some variety in the initial ice volume and thickness in the coupled simulations since these are uncertain but can have an impact on the evolution of ice sheets due to the hysteresis (by making

the climate slightly colder, e.g. Abe-Ouchi et al. 2013). Running a spin-up with BISICLES with different magnitude of SMB allowed us to implement this (L269-271).

**Comments about figures:**
**Figure 11**: I really liked this figure to try and understand how the different criteria used in the study relate. Is this all 200 ensemble members? What I find interesting, which I hope the authors can comment on is in panel (a) the same ice sheet volume, ~ 70m is produced for a GMST between 5C and 12C. Has the ice sheet not thicken? Changed in extent? I am trying to understand the 3 factors together. In terms of the southern margin, about ~ 11 C the southern margin has undergone a large retreat. Perhaps if the authors plot North American volume vs ice sheet margin this will become apparent.

Thank you! The figure includes 87 members satisfying the GMST constraint and Fig. R2 shows the relation between the southern extent and volume of the North American ice sheets. The result shows more variety in ice extent once the ice sheet volume exceeds 80m SLE.

It is very interesting to see similar ice volumes under different GMST. We have created a figure comparing the shape of the ice sheet within a particular ice volume range (80m - 90m SLE). The result implied a thicker but narrower ice sheet under warmer condition, but thinner and wider ice at colder conditions. This is implies some control from GMST on the shape of the ice sheet and is consistent with Fig. R2 showing a larger variety in the simulated ice extent beyond the ice volume of 80m SLE. However, we also need to be aware that these differences can be caused by differences in albedo and other parameters (Fig. S4). In this regard, we think further analysis is necessary to make this argument in the current paper. We might write a follow-up paper on this point, though, so thanks for the comment!

[Figure]

Fig. R2 Relationship of the North American ice volume and Southern extent at year 5000 of FAMOUS-BISICLES coupled simulations.

**Figure S3:** This is an interesting figure and from my understanding this is after the spin-up procedure (ice sheet only parameters)? If this is the case, the ice volume can reduce by up to 40m? Given than in the ice sheet-only stage (Fig S1) the volume in some simulation increases by ~ 20m, does this climate influence (feedbacks?) reduce this? This possible relates to my above question about spin-up, why not spin up with the climate feedbacks?

Fig. S3 shows responses of ice sheet in the first 500 years after the coupling of FAMOUS-BISICLES. The figure does show members with a reduction of ice volume by up to 40m SLE in the first 500 years. This is largely caused by the combinations of parameters producing very low albedo values, which result in as very large negative SMB. Having a larger ice sheet at the beginning of the coupling can induce a colder climate due to the cooling by the ice sheet itself. This may reduce the initial ice sheet melt, even if the albedo is low. However, Fig. S3 shows a very small impact from the initial ice volume on the ice sheet mass loss in the first 500 years. This means that, even starting from a larger ice sheet, the ice sheet can melt drastically if the albedo value is set to be low. We will add a following sentence to clarify this point;

"This suggests only a weak connection between final ice sheet volume at 5000 years and its initial volume at the beginning of the coupled simulations. (Similar results are also obtained for ice volume changes in the first 500 years.)".

**Figures changes:** The figures with multiple panels are small for the reader to see. This might be the typesetting of the manuscript but can the authors try to increase.

Done!

**Figure1:** I would suggest changing the title to a more general phrase. I am confused how there are SST across the land region? Is it SAT?

Masked out the values on land since it is SST.

**Figure2:** Can you add a key onto the figure to state: light blue = GMST; dark blue ...

Done!

**Figure 3**: Can you highlight the edge of the actual simulated ice sheet? It is hard to identify where the edge of the ice sheet is (panelsb,c,d,e) without guessing in reference to the ablation area. Does this figure only show grounded ice?

Done! The blue shades show the ice thickness, therefore it does contain some floating ice.

**Figures 4 and 7**: For these graphs can you add on the limit of GMST and ice volume as you have, for example on Figure 5.

Done! For Fig. 7, we could not include the shade since the ice volume constraint is applied only on the North American ice sheet, while the Figure shows the ice volume evolution of North America and Greenland.

**Minor comments**

Line 217: Laurentide> this is one ice sheet which makes up the LGM North American ice sheet: change to North American

Done!

**Terminology:** Can the author clarify from the beginning the difference between FAMOUS- ICE (is this with always an ice sheet? Or just the climate component): FAMOUS-Ice (Gandy et al., 2023) - this is when it is coupled to Glimmer, and FAMOUS-ICE, which then is referred to in the abstract as FAMOUS-BISICLES.
**GMT -** this is a very common abbreviation for other things: please change to GMST, Sat or something else.

Clarified the difference between FAMOUS-Ice and FAMOUS-BISICLES in the method section!

Changed GMT into GMST!